# Structure of a Rhs effector clade domain provides mechanistic insights into type VI secretion system toxin delivery

Brooke K. Hayes [1,2,5], Marina Harper [1,2,5], Hariprasad Venugopal[3], Jessica M. Lewis [1,2], Amy Wright[1,2], Han-Chung Lee[4], Joel R. Steele [4], David L. Steer [4], Ralf B. Schittenhelm[4], John D. Boyce [1,2] ✉ & Sheena McGowan [1,2] ✉

The type VI secretion system (T6SS) is a molecular machine utilised by many Gram-negative bacteria to deliver antibacterial toxins into adjacent cells. Here we present the structure of Tse15, a T6SS Rhs effector from the nosocomial pathogen *Acinetobacter baumannii*. Tse15 forms a triple layered β-cocoon Rhs domain with an N-terminal α-helical clade domain and an unfolded C-terminal toxin domain inside the Rhs cage. Tse15 is cleaved into three domains, through independent auto-cleavage events involving aspartyl protease activity for toxin self-cleavage and a nucleophilic glutamic acid for N-terminal clade cleavage. Proteomic analyses identified that significantly more peptides from the N-terminal clade and toxin domains were secreted than from the Rhs cage, suggesting toxin delivery often occurs without the cage. We propose the clade domain acts as an internal chaperone to mediate toxin tethering to the T6SS machinery. Conservation of the clade domain in other Gram-negative bacteria suggests this may be a common mechanism for delivery.

Bacteria live in complex environments where nutrient availability is often poor. In order to compete with surrounding bacteria for space and resources, ~25% of Gram-negative bacteria utilise a Type VI Secretion System (T6SS)[1]. Functionally similar to an inverted T4 bacteriophage tail, this secretion system delivers effector proteins[2] directly into either eukaryotic or prokaryotic cells[3]. While eukaryotic effectors generally act to manipulate the host cytoskeleton or are involved in evasion of host defences[4–6], prokaryotic effectors usually act to kill prey bacteria by targeting essential cell structures, such as nucleic acids, peptidoglycan or the cell membrane[7,8]. To prevent self- and sibling-killing, the predator bacterium expresses immunity proteins that bind and neutralise their cognate toxic effector[9–12].

T6SS effectors are delivered via interaction with one of the three T6SS needle/tip structural proteins; PAAR, VgrG (TssI)[13] or Hcp (TssD)[14]. These effectors may be classified as specialised or cognate effectors. Specialised effectors have the effector domain translationally fused to one of the T6SS structural proteins, while cargo effectors are encoded independently of their cognate T6SS protein and interact through non-covalent interactions[8,15,16]. Chaperones may also be required to permit these interactions[17–19].

The nosocomial Gram-negative pathogen *Acinetobacter baumannii* is known to utilise a T6SS system for bacterial competition[20]. Often infecting the critically ill, *A. baumannii* accounts for up to 21% of hospital-acquired infections in intensive care units[21]. Clinical treatment of *A. baumannii* infections are challenging due to extensive and widespread drug resistance. As such, *A. baumannii* is rated as a priority 1 critical pathogen and therefore new approaches to infection control are urgently required[22].

[1]Biomedicine Discovery Institute, Department of Microbiology, Monash University, Clayton, VIC, Australia. [2]Centre to Impact AMR, Monash University, Clayton, VIC, Australia. [3]Ramaciotti Centre for Cryo-Electron Microscopy, Monash University, Clayton, VIC, Australia. [4]Monash Proteomics & Metabolomics Platform, Biomedicine Discovery Institute, Monash University, Clayton, VIC, Australia. [5]These authors contributed equally: Brooke K. Hayes, Marina Harper. ✉e-mail: John.Boyce@monash.edu; Sheena.McGowan@monash.edu

We recently characterised the T6SS effector and immunity protein pairs from the *A. baumannii* clinical isolate AB307-0294[23]. Three effector/immunity pairs were identified, including one pair designated Type VI secreted effector 15 (Tse15; originally Rhs1) and Type VI secreted immunity 15 (Tsi15; originally Rhs1I)[23,24]. Bioinformatic analyses of Tse15 indicated that it is a cargo effector belonging to the rearrangement hotspot (Rhs) family[23]. However, the toxic domain showed no amino acid similarity to characterised T6SS toxins. Generally, Rhs-family proteins are distinguished by an N-terminal domain, central Rhs domain comprising YD repeats and a toxic variable C-terminal domain (CTD)[25]. The conserved cleavage motif 'DP(I/L) GXXGGX$_5$YX$_8$D(P/S)XG(L/W)' is found between the Rhs domain and toxic CTD[24,25]. Cleavage at this site is proposed to release the CTD away from the Rhs core and is required for CTD toxic activity[26,27]. It is widely accepted that effector release is mediated by aspartyl protease self-cleavage, although what stimulates this during or following delivery is unknown. Preliminary functional analysis of the Tse15 CTD alone (Tse15tox) showed that cytoplasmic expression of Tse15tox was able to kill *E. coli* and that Tse15tox toxicity was inhibited by expression of the predicted immunity protein, Tsi15[23].

T6SS Rhs-family effectors are found in several Gram-negative pathogens[28]. For many years, the roles of the clade and Rhs domains were largely unknown, with a general hypothesis that the Rhs domain physically shields the toxin to protect the predator bacteria[29,30]. Recently, the toxic CTD of a *Photorhabdus laumondii* T6SS Rhs effector was shown to be constrained within the cocoon or cage-like structure formed by the Rhs domain, confirming its likely role in physical shielding[31].

Here we present the structure of *A. baumannii* Tse15 and show that Tse15 also comprises a β-cocoon like structure and map unfolded toxin density enclosed within the Rhs cage. We also map the region of VgrG15 that interacts with the N-terminal region of Tse15 for delivery by the T6SS and provide a model for toxin delivery via the T6SS. We show that the Rhs cage domain is unlikely to be delivered outside of the cell, suggesting that the N-terminal clade domain pulls the Tse15-tox domain out of the cage during delivery. This would indicate that toxic activity is activated either during or after delivery. We predict that this may be a common mechanism of Rhs effector toxin delivery and activation. Our findings help to elucidate the complex molecular mechanisms by which toxic effectors are delivered by the T6SS and identify potential targets for disrupting delivery by VgrG bound cargo effectors.

## Results

### Tse15 contains a Rhs repeat β-cocoon domain and an α-helical clade domain

Bioinformatic analysis of Tse15 indicated that, like other Rhs effectors, the protein likely comprises three domains, an N-terminal clade-specific domain, a core Rhs domain and a C-terminal domain (CTD) with toxic activity[23] (Fig. 1a). Expression of recombinant full-length wildtype Tse15 in *E. coli* produced soluble protein that resolved as a large single species on analytical size-exclusion chromatography (Fig. 1b). Similar to other Rhs proteins, resolution of purified Tse15 on SDS-PAGE showed three bands (Fig. 1c). Western blotting (Fig. 1c below), mass spectrometry-based peptide fingerprinting and N-terminal sequencing (Supplementary Fig. 1) confirmed that the three fragments corresponded to the predicted N-terminal clade domain (residue 1-334); the central Rhs domain (residues 335-1395) and the toxic CTD (residues 1396-1590). As only a single peak was observed following analytical size-exclusion, and all three domains were co-purified by nickel purification (despite only the CTD containing a His-tag), we concluded that the three domains remain tightly associated in solution following cleavage.

Single particle cryo-electron microscopy (cryoEM) of the purified wildtype Tse15 revealed two major 3D particle classes; both

monomeric in assembly but one that resembled the expected β-cocoon of the Rhs domain and a second that had the Rhs domain with a globular protrusion at one end (Supplementary Fig. 2, purple and yellow respectively). To understand the nature of the particle classes and to assist with model building, we used AlphaFold2 to produce full length (residues 2–1590) Tse15 models that were compared to the particles and fitted to the resulting maps. The top ranked model indicated with high confidence that the Tse15 Rhs domain formed the expected large β-cocoon structure with the toxin present inside the cage (Supplementary Fig. 3a, b). The confidence scores for the toxin domain were low, and as such, we removed this region from the model coordinates. This model also suggested that the globular protrusion observed in one of the 3D particle classes was likely to be the clade domain predicted to comprise a bundle of α-helices (Supplementary Fig. 3a, orange). This provided visualisation of a Rhs-associated domain that has not been resolved in other similar Rhs effector structures[26,31,32].

A map of Tse15 was produced with the clade domain present at a maximum resolution of 3.08 Å (Supplementary Fig. 2, Supplementary Table 1, Fig. 2a). The top ranked AlphaFold2 model without the toxin

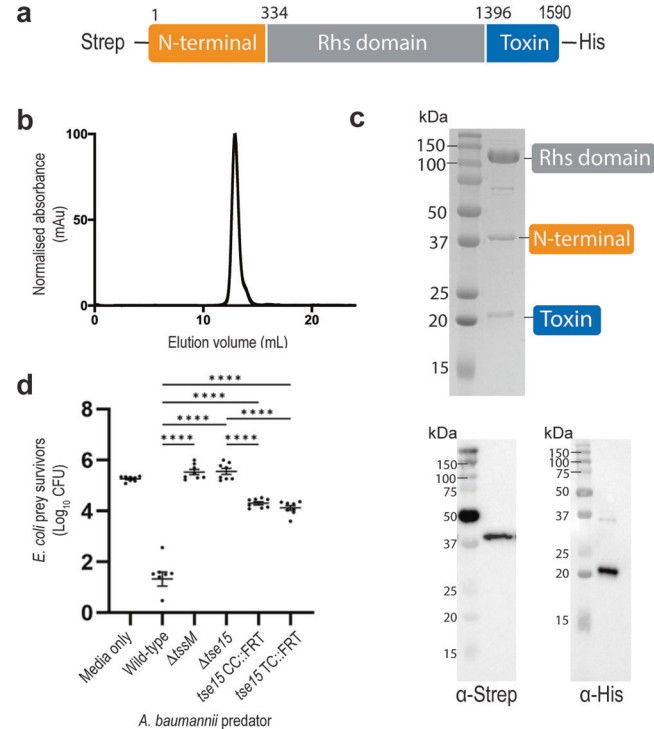

**Fig. 1 | Tse15 separates into three separate fragments that stay tightly associated during protein expression and purification. a** Construct design for production of Tse15; N-terminal clade domain is orange, Rhs domain is grey and toxin domain is blue. Purification tags and domain boundary residue numbers are indicated. **b** Analytical size-exclusion chromatogram showing that Tse15 elutes as a single peak. **c** Coomassie stained SDS-PAGE gel of purified Tse15 and domains associated with each band. Molecular weight markers (kDa) are shown on left hand side of gel. (Below) Western blots showing domain separation during purification probed with α-Strep and α-His antisera (as indicated, *n* = 1). Molecular weight markers (kDa) are shown on left hand side of blots. **d** T6SS competitive killing assays to measure the effect of replacing the Tse15 clade cleavage motif or the toxin cleavage motif with an in-frame FRT site on the ability of *A. baumannii* to kill vulnerable *E. coli* prey in a Tse15-dependent manner. Predator strains used were AB307_0294 wild-type, a Δ*tssM* mutant (inactive T6SS), a Δ*tse15* mutant, a *tse15* clade cleavage mutant (*tse15*CC::FRT) and a *tse15* toxin cleavage mutant (*tse15*TC::FRT). Bars represent mean of four biological replicates, error bars represent SEM. Statistical significance was determined using ANOVA with Tukey's multiple comparisons test. ****$p < 0.0001$.

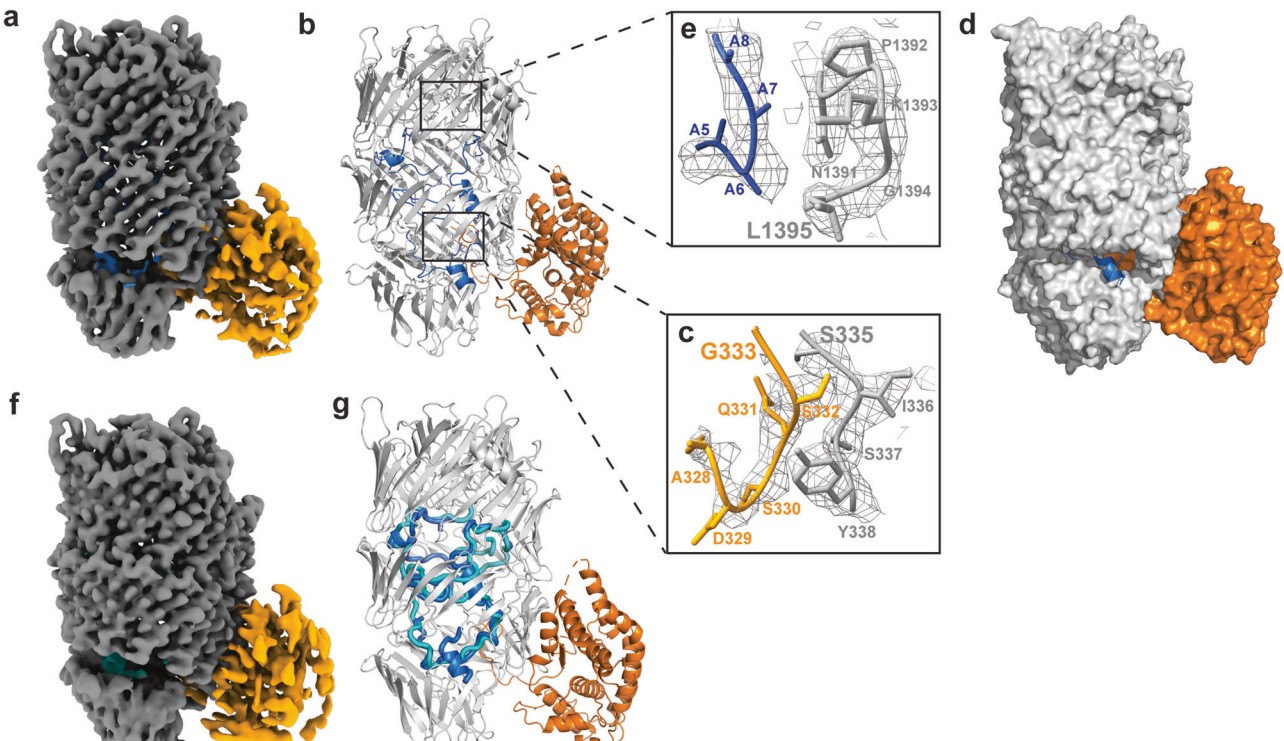

**Fig. 2 | Single particle cryoEM structures of Tse15. a** Tse15 wild-type density map at a threshold of 0.75 (2.85 RMSD) where the N-terminal domain is coloured orange, Rhs cage grey and the toxin peptide(s) in blue. **b** Cartoon depiction of Tse15 structure coloured by domain (clade orange, Rhs grey and toxin peptides blue). Four separate unsequenced toxin peptides (106 of 195 residue CTD domain) could be modelled into the density map. To the right, Tse15 density map as mesh (ChimeraX) volume viewer at a contour of 6.45 RMSD showing fit of (**c**) clade-Rhs autocleavage site (S335 in grey; G333 in orange) and (**e**) toxin cleavage site (L1395, in grey) with toxin peptide shown in blue and Rhs domain in grey. Residues numbers are indicated. **d** Surface depiction of Tse15 clade and Rhs domain coloured orange and grey respectively, toxin is shown as blue cartoon. **f** Tse15$_{NN}$ density map at a threshold of 0.15 (1.2 RMSD) where the N-terminal domain is coloured orange, Rhs cage light grey and the toxin domain cyan. **g** Cartoon depiction of aligned Tse15 and Tse15$_{NN}$ where peptide toxin density is shown in blue for Tse15 and cyan for Tse15$_{NN}$.

domain (residues 1–1395) was fitted to the map using UCSF Chimera fitmap. Following refinement and manual adjustments, the final model of Tse15 spanning residues 14–1395 was produced that resolved the structure of an α-helical clade domain (residues 14–333) and a core β-strand rich cage (residues 335–1395) (Fig. 2b, in orange and grey respectively). The clade domain was shown to consist of a bundle of 12 α-helices (Fig. 2b). Some loop regions (residues 190–203; 288–297) on the exterior of the clade domain could not be modelled, and overall, resolution of this domain was weaker than the Rhs domain. We could resolve the cleavage point between the N-terminal clade and Rhs domains, with the sequence matching the previously identified cleavage site and a clear break in the connective density of the polypeptide chain (Fig. 2c). Interrogation of clade domain fold using both DALI[33] and FoldSeek[34], showed that it lacked homology to any experimentally resolved protein structures. However, we were able to match the fold to numerous AlphaFold2 database predictions, identifying many Rhs-containing proteins from Gram-negative bacteria including *Pseudomonas* spp. and *Burkholderia* spp. (Table S2, Supplementary Fig. 4). This shows that the fold is common across a range of Gram-negative pathogens, and we show experimental confirmation of the fold.

The central Rhs domain was comprised of β-strands arranged in antiparallel β-sheets that twist into three sub-structures to form a hollow cage that is ~55 Å wide and ~100 Å long. We were able to confidently model all residues except 1256-1267 that appeared to form a solvent exposed exterior loop that lacked density. The Tse15 Rhs domain exhibited significant structural similarity to other bacterial Rhs structures (7.5 Å RMSD to 7PQ5, 5.4 Å RMSD to 7Q97, 4.6 Å RMSD to 8H8A and 14.0 Å to 8H8B)[26,31,32], as well as to other YD repeat proteins, including Tc toxins (14.1 Å RMSD to 4IGL)[29] and eukaryotic teneurins

(19.0 Å RMSD to 6FB3)[35]. Inspection of the surface of Tse15 showed a large cleft-like opening to the interior of the protein complex that was formed between the first and second β-cocoon sub-structures (Fig. 2d, Supplementary Fig. 5a). The opening is asymmetrical, located on only one side of the cage and is close to the clade domain. A small hole measuring ~10 Å in diameter is also present close to the larger opening and is the only other access point to the interior (Supplementary Fig. 5b). Both the top and bottom of the Rhs cage are effectively plugged and offer no access to the interior. Through the large entrance, the clade autocleavage position (residue S335) can be seen to be well coordinated inside the cage (Supplementary Fig. 5c). The interaction surface between the clade and Rhs domain was long, spanning eight β-strands of the Rhs cage (~28 Å) and had a buried surface area of 2396 Å. Interrogation of the residue interactions between the clade and Rhs domain showed a total of 19 hydrogen bonds and 15 salt bridges (Supplementary Table 3, Supp Fig. 5d). Due to the spread of these residues, we did not pursue a mutagenesis approach to verify the interaction surface as individual amino acid mutations were unlikely to disrupt the interface, while more significant internal deletions were likely to disrupt the overall fold of either domain. The high number of potential salt-bridges in the interface does, however, suggest that dissociation of the clade would likely be pH labile.

After fitting of the clade and Rhs domains (A-chain), a difference map was generated using UCSF Chimera (v1.14)[36] which allowed us to produce a map of any unmodelled density. This map clearly showed peptide elements within the β−stranded cage (Supplementary Fig. 6a). Given the only atoms left to be modelled were the toxic CTD domain, we concluded that this was the toxin enclosed within the cage.

The toxin density could be observed lining the interior wall of the β−stranded cage and appeared to have elements of secondary structure but not an overall globular fold. Between the Rhs and the toxin, there is a clear break in connective density that corresponds to the cleavage site of the toxin as determined by N-terminal sequencing (L1395, Fig. 2e, Supplementary Fig. 7a). The break in backbone density meant that we were not able to pinpoint the first residue of the toxin (N1396) and coupled with the resolution of the data, were unable to manually build the sequenced structure. We were able to de novo build four separate peptides (chains B, C, D and E) into regions of convincing density, placing a total of 106 of the 195 residue CTD toxin (Fig. 2b, blue). These chains were modelled as poly-alanine with some glycine residues placed to allow an improved fit to the density. Given the high overall sequence percentage of glycine in the toxin CTD (14.6 %), we felt modelling glycine was appropriate. To attempt to trace the toxin sequence within the modelled peptide(s), we also performed cross-linking mass spectrometry using Tse15. Unfortunately, no peptide crosslinks were observed between the clade or Rhs domain and the toxin, providing no anchor point to map the toxin sequence to structure (Supplementary Fig. 8a, b). The lack of inter-domain toxin crosslinking could be the result of an absence of lysine residues to allow for crosslinking but may also suggest there is disordered interaction, further supporting our finding that the toxin is unfolded within the cage.

The toxin peptides modelled within the cage showed a strong tendency to interact with the interior wall of the Rhs β-strand substructures. Similar toxin and Rhs cage interactions have also been observed in an Rhs nuclease toxin from *V. parahaemolyticus*[26]. We were unable to define any peptide in the middle of the cage void. We also noted that the short helical chain C (10 residues) was localised to the exterior cleft of the Rhs domain, suggesting that this is indeed the point of toxin entrance or exit (Fig. 2d, blue). Also localised close to the entrance and chain C was the longer chain E (43 residues) that appeared to interact with the end of the clade domain, close to the cleavage site (Supplementary Fig. 5c). The position of the toxin is intriguing and points to a possible role for the clade in toxin release.

### Mapping the toxin inside the Rhs cage

We reasoned that if we could block autocleavage of the toxin within Tse15 we would be able to map the toxin sequence within the cage guided by continuous chain density. Previous studies have shown that Rhs effectors possess aspartyl protease activity that drives auto-cleavage of the toxin[26,27,31,32]. Two aspartic acid residues are essential for this cleavage mechanism. We mutated the equivalent two aspartic acid residues (D1369N, D1391N) to asparagine in a single construct (Tse15$_{NN}$) (Supplementary Fig. 9a). SDS-PAGE analysis of the purified Tse15$_{NN}$ showed only two protein fragments, compared to the three observed during purification of wildtype Tse15. Indeed, the 22 kDa toxin CTD observed in the Tse15 purification was absent in the Tse15$_{NN}$ purification and the Rhs domain fragment showed a corresponding increase in size (Supplementary Fig. 9b). The identity of this toxin protein fragment was confirmed by mass spectrometry peptide fingerprinting (Supplementary Fig. 9c), thus confirming that wildtype Tse15 possesses aspartyl protease activity that drives self-cleavage of the toxin from the Rhs domain.

Unlike wildtype Tse15, particle processing for the cryo-EM data of Tse15$_{NN}$ showed only one main monomeric 3D particle class with the clade domain predicted to be retained together with the body of the Rhs cage (Supplementary Fig. 10). Tse15$_{NN}$ particles were compiled to produce a map to a maximum resolution of 1.77 Å (Supplementary Fig. 10, Supplementary Table 1, Fig. 2f) and showed that the Tse15$_{NN}$ quaternary structure is the same as the wildtype (RMSD 0.860 Å over residues 14–1395, A chain). Clear, connective density was observed after L1395 and the sequence of the toxin could be mapped up to residue 1484, guided by a difference map as well as the unsharpened map (Supplementary Fig. 6b, Supplementary Fig. 7b). After residue 1484, significant breaks in connective density meant that the C-terminal of the toxin could not be placed into the maps. Overall, the position of the Tse15$_{NN}$ toxin was similar but not the same as the wildtype Tse15, and lacked defined secondary structure features (Fig. 2g). We also could not find equivalent density for the helix that was present in the Rhs domain opening but did show the toxin peptide was still in close proximity to the end of the clade domain. In common, however, was the disordered nature of the toxin and its propensity to interact or line the interior wall of the Rhs domain. This result also suggests that tethering the toxin to the Rhs domain (preventing autocleavage) may influence the position of the toxin in the cage, a likely outcome given our result that showed no true position of the toxin in solution. From the two structures, it appears clear that the toxin must leave the cage prior to achieving its final folded and active conformation.

### Autocleavage of the clade and toxin are independent events

Our purification of either wildtype Tse15 or Tse15$_{NN}$ showed that in both cases the clade domain was cleaved from the polypeptide chain; we also confirmed this in our structural data for Tse15 as we could definitively identify the cleavage position. To understand if this event may be common to other *Acinetobacter* Rhs effectors, we aligned a representative sequence from groups 5, 6, 15, 16, 21, 22, 26 and 27 (as classified by Lewis et al.[24]) with our experimentally determined cleavage site and assigned the cleavage site as position 0. Inspection of the alignment surrounding position 0 identified that residues at +2, +7, +9, +13 and +15 were completely conserved across all groups assessed (Supplementary Fig. 11a). Additionally, high amino acid similarity was observed in the other positions. Taken together, we identified a conserved motif of (S/N)Ix$_4$G(T/A)Ex$_3$HxD where the S/N (S335 as per Tse15 numbering) is the P1′ residue (C-terminal to the scissile bond). We were interested to know if this motif was conserved beyond the *Acinetobacter* genus, so we again probed FoldSeek[34], using the same search criteria that we applied to identify the motif, which included 19 residues beyond the cleavage site. The motif was highly conserved (Supplementary Table 4), suggesting that it is common to this family of Rhs effectors. Of the 56 sequences that showed structural similarity, approximately eight effectors did not contain the consensus sequence as the structure did not go beyond the N-terminal domain. As the N-terminal domain can be identified without a Rhs domain and effector, this suggests a possible role for the N-terminal clade domain beyond T6SS binding.

Structurally, we located this motif between two interior back-to-back β-sheets in the bottom substructure (Supplementary Fig. 11b). Recent studies of Rhs effectors from *Aeromonas dhakensis* and *Vibrio parahaemolyticus* showed that mutation of a glutamic acid residue in proximity to the N-terminal cleavage site, prevented autocleavage[26,27]. The motif in the *Acinetobacter* Rhs effectors contained a single conserved glutamic acid (E343 Tse15 numbering). Our structures show that although E343 can form a hydrogen bond with the backbone nitrogen of S335, it is more likely that a salt bridge may be formed between K334 and E343. Our structures did not resolve K334 but the AlphaFold2 model used as the initial structure indicates that the two sidechains are an appropriate distance for such an interaction (Supplementary Fig. 11b). To test this hypothesis, we generated Tse15$_{E343A}$ and Tse15$_{K334A,S335A}$ (Tse15$_{KS}$) to assess the importance of the specificity of the cleavage site. The Tse15$_{KS}$ mutant showed reduced cleavage of the N-terminal clade domain compared to wildtype, suggesting that K334 and S335 are important but not essential for cleavage (Supplementary Fig. 11c). However, cleavage of the N-terminal clade domain was completely inhibited in the Tse15$_{E343A}$ mutant, suggesting E343 acts as the essential catalytic nucleophile for autocleavage. Interestingly, the Tse15$_{E343A}$ mutant still underwent autocleavage of the toxin, confirming the two cleavage events are independent of each other.

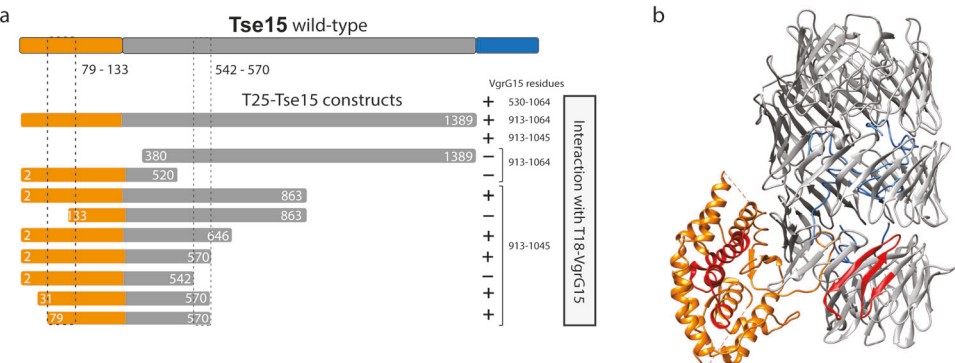

**Fig. 3 | *E. coli* two-hybrid analysis of various N-terminal truncations of Tse15Δtox with VgrG15. a** Schematic representation of the regions of Tse15 fused to the C-terminal end of the T25 adenylate cyclase fragment. Each was tested for their ability to interact with one or more regions of VgrG15 fused to the C-terminal end of the T18 adenylate cyclase fragment (text at right). Region of Tse15 and derivatives shaded orange represents the clade domain, grey regions represent the Rhs domain and blue region (in Tse15 wildtype only) represents the toxin. Numbers indicate amino acid residues of wildtype Tse15 or VgrG15 each fragment represents. Interaction between a T25 fusion proteins and a T18 fusion protein is indicated with a +, no interaction is indicated by a dash. An example of a positive and negative colony can be seen in Fig. 4c. **b** Identified interaction sites mapped onto Tse15, with the clade domain shown in orange, the Rhs domain in grey, toxin domain in blue and the interacting regions shown in red.

To address the biological importance of the two cleavage motifs (clade and toxin) in Tse15, we constructed two marker-less mutants in *A. baumannii* strain AB307-0294 using an in-frame flippase recognition target (FRT) approach. The two mutant strains, *tse15*CC::FRT (AL4734) and *tse15*TC::FRT (AL4745) had the clade cleavage motif (Tse15 amino acids 335-346) or the toxin cleavage region (amino acids 1384-1395) replaced with an in-frame FRT site, respectively. In competition with *E. coli* prey, the Tse15 cleavage mutant strains grew at rates indistinguishable to the growth of wild-type AB307-0294 (Supplementary Fig. 11d). Importantly, both had a significantly reduced ability to kill *E. coli* prey vulnerable only to Tse15-mediated killing (Fig. 1d). This result indicates that both cleavage sites are important for the proper function of the Tse15 toxin.

### Tse15 appears to interact with VgrG15 using an edge-to-edge β-strand contact mechanism

Rhs cargo effectors must interact with their cognate VgrG protein, located on the tip of the T6SS complex, for delivery into prey cells[13]. In *A. baumannii* strain AB307-0294, delivery of Tse15 into *E. coli* prey cells requires a functional VgrG15 protein[23]. This was initially determined by Fitzsimons et al. and was experimentally confirmed in our study by a pulldown of native VgrG15 by purified Tse15 (Supplementary Fig. 12)[23]. To map the specifics of the interaction between VgrG15 and Tse15, we conducted *E. coli* two-hybrid analysis using sub-fragments representing the variable C-terminal region of VgrG15 together with Tse15 sub-fragments with sequential deletions from the N-terminus (Fig. 3a). As our structural data suggested the toxin remained within the Rhs cage until deployment, and to avoid toxicity issues, we chose to use fragments without the CTD (Tse15$_{\Delta tox}$) as the base for these experiments. We identified two regions that were required for the interaction of Tse15$_{\Delta tox}$ with VgrG15. The minimal interacting fragment was from residues 79-570, with deletions from either end abrogating the interaction. Thus, this suggests that amino acids between 79-133 and 542-570 of Tse15$_{\Delta tox}$ are crucial for the interaction (Fig. 3a, b) and that sections of both the N-terminal clade and Rhs domains of Tse15 are important for the interaction with VgrG15.

To determine the regions of the VgrG15 required for interaction with Tse15, we first inspected the VgrG15 sequence and homologous structures. Alignment of VgrG15 with the two other *A. baumannii* AB307-0294 VgrGs (VgrG16 and VgrG17) revealed the N-terminal region of all three VgrGs was highly conserved, while the C-terminal region was diverse (Supplementary Fig. 13). Mapping this information to the known structure of a VgrG1 homotrimer from *P. protegens*, indicated that the conserved N-terminal domain was involved in the inter-collating trimeric stalk structure but there was little to no structural information on the C-terminal domains of this or other VgrG proteins[32]. We concluded therefore that the N-terminal region of VgrGs are likely to be responsible for formation of the central spike and for the interaction with the T6SS machinery and the C-terminal variable regions are likely to be responsible for interaction with their cognate effectors[37]. To confirm this, a chimeric VgrG protein was produced. The chimera combined the N-terminal region of the AB307-0294 VgrG16, which is normally involved in delivery of the DNase effector Tde16 (VgrG16$_{1-833}$), with the VgrG15$_{831-1064}$ C-terminal region[23] (Fig. 4a). The chimera was introduced into an *A. baumannii* Δ*vgr15* mutant and the ability of the chimera to complement this strain for Tse15 delivery was measured by competitive killing assays. These assays were conducted with *E. coli* prey that expressed the immunity proteins Tdi16 and Tai17 in *trans*, thus this strain was only vulnerable to the activity of Tse15. As delivery of Tse15 into prey is dependent on a functional VgrG15 protein, any Tse15-mediated killing of *E. coli* could be attributed to the chimeric protein non-covalently interacting with Tse15 (Fig. 4b). As a control, the chimera was also introduced into an *A. baumannii* Δ*vgrG16* mutant and its ability to kill *E. coli* prey vulnerable only to the activity of Tde16 assessed. The VgrG chimera was able to function as interaction partner for Tse15 but not for Tde16. This showed that VgrG15 residues 831-1064 were required for interaction with Tse15, confirming that the specific effector delivery determinants are within the C-terminus.

We next constructed a series of VgrG15 C-terminal truncations and assessed each for their ability to facilitate Tse15-mediated toxicity. Killing assays using *E. coli* prey vulnerable only to Tse15-mediated killing showed that the last 54 amino acids of VgrG15 were not required for interaction with Tse15 (Fig. 4b). VgrG15 proteins with C-terminal truncations between 84 and 114 residues displayed an intermediate level of killing while any truncations beyond the last 152 amino acids resulted in a non-functional VgrG15 that could not deliver Tse15 (Fig. 4b). To confirm a direct interaction between the two proteins, we used *E. coli* two-hybrid analysis to look for interaction between the C-terminal fragment of VgrG15 (amino acids 530-1064) and Tse15$_{\Delta tox}$, translationally coupled to the C-terminal end of the adenylate cyclase fragments, T18 and T25, respectively. A T18 fragment fused to VgrG15 amino acids 913 – 1064 showed a strong interaction with T25 fused to Tse15$_{\Delta tox}$, strongly supporting our assays with the VgrG16:VgrG15 chimera and truncated VgrG15 derivatives (Fig. 4c). We made four further constructs that removed different residues from both termini

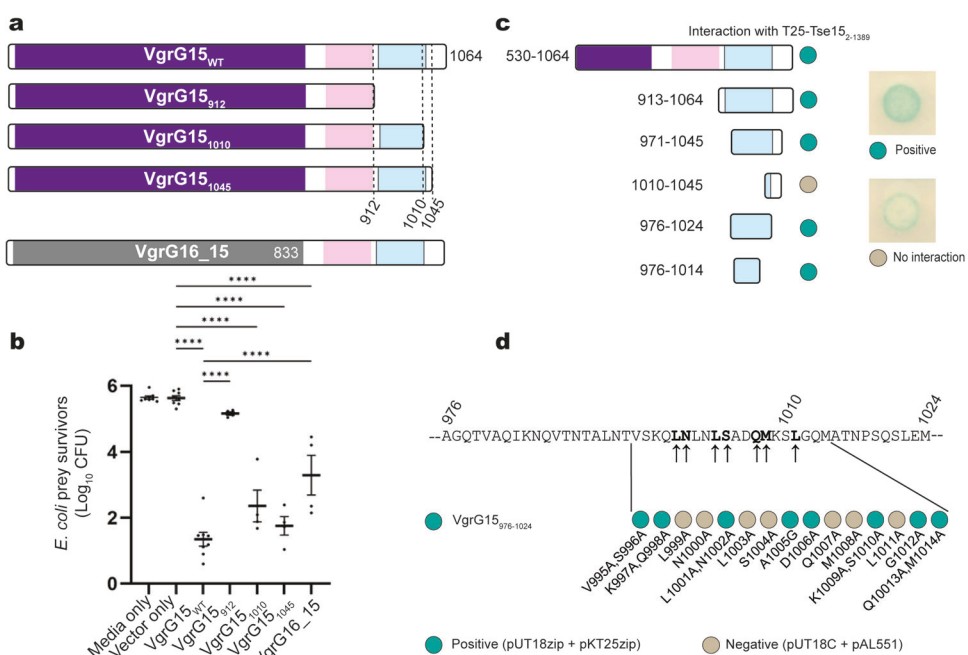

**Fig. 4 | Mapping the VgrG15 interactions with Tse15 in vivo and in vitro.**
**a** Schematic representation of AB307-0294 wild-type VgrG15, deletion derivatives and a VgrG16:15 chimera. The purple and grey shaded area indicates conserved VgrG15 and VgrG16 domains respectively. Pink shaded region represents the VgrG15 Ig-like domain (residues 853-907) and blue shaded region indicates region predicted to form five α-helices (VgrG15 residues 933-1028). **b** T6SS competitive killing assays were used to determine the region of VgrG15 required for Tse15 delivery into *E. coli* prey cells (susceptible only to Tse15-mediated killing). The predator *vgrG15* mutant was provided in *trans* with wild-type *vgrG15*, truncated *vgrG15* constructs (subscript numbers indicate encoded VgrG15 amino acids), or a gene encoding a chimera VgrG16_15$_{831-1064}$ protein. Bars represent mean of four biological replicates, error bars represent SEM. Statistical significance was determined using ANOVA with Tukey's multiple comparisons test. ****$p < 0.0001$.
**c** Bacterial adenylate cyclase two-hybrid analysis to measure the direct interaction of various regions of VgrG15 (fused to the C-terminal end of the T18 fragment of

adenylate cyclase) with Tse15$_{\Delta tox}$ (fused to the C-terminal end of the T25 fragment of adenylate cyclase). Positive interaction between the two fusion proteins is indicated as blue/green circle while lack of interaction is indicated as cream circle (shown right of panel). **d** Alanine scanning of smallest region of VgrG15 still able to interact with Tse15$_{\Delta tox}$. The ability of each alanine substituted protein (T18-VgrG$_{976-1024}$ parent) to interact with Tse15$_{\Delta tox}$ was assessed using the bacterial adenylate cyclase two-hybrid system. Arrows indicate the amino acids (bold) that when substituted with alanine resulted in a failure of T18-VgrG$_{976-1024}$ to interact with T25-Tse15$_{\Delta tox}$. Representative images (minimum of three biological replicates) shown for interactions of Tse15$_{\Delta tox}$ with single and double alanine substitutions in the VgrG region of interest. Blue/green circles indicate the two recombinant proteins co-expressed are interacting. Cream coloured circles indicate no interaction was observed, showing the substituted amino acid(s) may be involved in the interaction between the two proteins.

(Fig. 4c). These two-hybrid data indicated that the minimal region of VgrG15 required for interaction with Tse15$_{\Delta tox}$ was between residues 976 and 1014 (Fig. 4c). Examples of positive and negative interactions in the bacterial two-hybrid assays are shown (Fig. 4c, right).

To further assess VgrG15 residues crucial for interaction with Tse15, we undertook alanine scanning of the region between residues 976 and 1024 using the bacterial two-hybrid system. The results identified residues L999, N1000, L1003, S1004, Q1007 and L1011 as required for the interaction (Fig. 4d, Supplementary Fig. 14). To determine if the alanine substitutions identified using two-hybrid analysis prevented interaction of VgrG15 with Tse15 in vivo in *A. baumannii*, we constructed an expression plasmid encoding full length VgrG15 with all seven alanine substitutions (L999A, N1000A, L1003A, S1004A, Q1007A, M1008A and L1011A; VgrG15$_{ala}$). The expression plasmid was introduced into the AB307-0294 Δ*vgrG15* mutant. As a control, pAL1415 encoding wild-type VgrG15 was separately used to transform the mutant. Surprisingly, *A. baumannii* expressing VgrG15$_{ala}$ in *trans* was still able to kill *E. coli* at levels similar to the mutant expressing wild-type VgrG15, suggesting that the region identified by two-hybrid analysis was not the sole mediator of the interaction between full length VgrG15 and Tse15 in *A. baumannii*. We note that all VgrG proteins used in the killing assays in *A. baumannii* retained the N-terminal region, allowing native trimerization of a VgrG15 T6SS spike. In contrast, the two-hybrid assays in *E. coli* used smaller regions of VgrG15 translationally coupled to the C-terminal end of an adenylate

cyclase fragment. Despite this difference, the two-hybrid analyses supported the competitive killing assays that indicated VgrG15 uses the region between amino acid 912 and 1010 to interact with its cognate effector Tse15.

To consider these interactions in relation to potential quaternary structure, we mapped our interaction data onto a model generated by AlphaFold2 multimer using the C-terminal domain of VgrG15 (851-1064) and full length Tse15 structure. These data allowed us to produce an overall binding model that shows Tse15 tethered to a VgrG15 homotrimer with an unusual interface that predominantly involved the Tse15 clade domain (Fig. 5a). The top ranked model (based on iPTM score) suggests that a loop within the Tse15 clade domain (residues 288-304, not defined in our structures) forms a β-strand that interacts with a five-stranded Ig-like domain within VgrG15 (residues 853-907) (Fig. 5b). This strand insertion would allow for a strong interaction between VgrG15 and Tse15, which would be necessary if VgrG15 is required to remain tethered to Tse15 as the T6SS fires. Importantly, our truncation data show that versions of VgrG15 that retain the residues to form this Ig-like fold, but not further C-terminal residues, cannot deliver the Tse15 effector (VgrG15$_{1-912}$; Fig. 4a). Following this edge-to-edge contact region, residues 908 - 932 of VgrG15 appear to cross the surface of the Tse15 N-terminal clade domain then form five α-helices (in residues 933-1028). These helices are located in approximately the same region of VgrG15 we identified as important for VgrG15:Tse15 interactions (Fig. 4). When aligning the alanine scanning bacterial two-

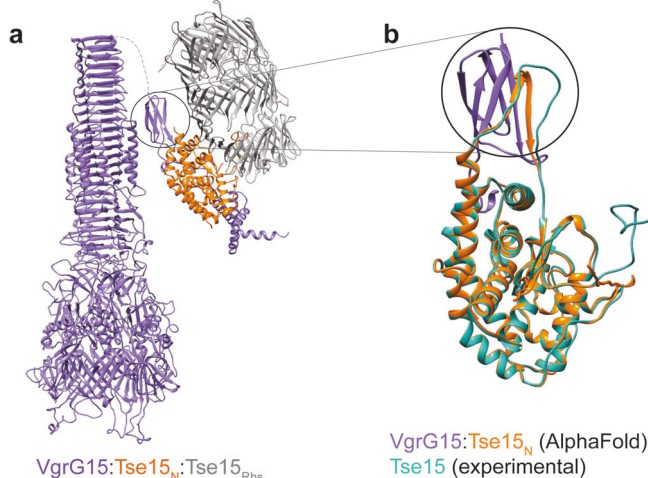

**Fig. 5 | Binding model for Tse15:VgrG15 prior to Rhs domain dissociation.**
**a** Model of the full length VgrG15 homotrimer in purple interacting with the Tse15 AlphaFold2 model coloured by domain (orange clade domain and grey Rhs domain). **b** AlphaFold2 model of VgrG15 residues 853-907 alone (purple) interacting with the Tse15 clade (residues 20-313, orange). The cryo-EM model of Tse15 clade domain is overlaid (teal). Black circle highlights a change in N-terminal clade domain secondary structure when involved in an edge-to-edge strand interaction with VgrG15.

VgrG15:Tse15_N:Tse15_Rhs

VgrG15:Tse15_N (AlphaFold)
Tse15 (experimental)

hybrid data to the model, the amino acids important for interaction between VgrG15 and Tse15, when fused to adenylate cyclase fragments, are clustered around a loop and helix region that sits below the clade domain. This suggests that the interactions that sit both on top and below the clade domain are important for the interaction between VgrG15 and Tse15. However, as Tse15-mediated killing of *E. coli* was observed when VgrG15_ala was used as the cognate VgrG15 protein, we propose that the edge-to-edge region at the Ig-like fold is also important for delivery of Tse15.

### The Rhs β-cocoon is rarely secreted out of the predator cell

Our structural data clearly show that the Tse15 toxin is retained in the Rhs β-cocoon, suggesting that the cocoon likely protects the host cell from Tse15 toxicity. Thus, we reasoned that the cognate immunity protein, Tsi15, is only required to prevent sister, but not self-killing. To test this, we constructed a double mutant that had both the T6SS system inactivated (via deletion of *tssM*) and the *tsi15* gene deleted. This mutant would produce Tse15 but be unable to export Tse15 due to T6SS inactivity and would lack the cognate immunity protein Tsi15 that neutralizes Tse15 toxicity. Growth curve analysis conducted in LB media showed that the Δ*tssM*Δ*tsi15* double mutant was as viable as wild-type *A. baumannii* AB307-0294 (Supplementary Fig. 15), confirming that Tsi15 is not required to protect the cell from self-intoxication with Tse15.

There is very limited data on how Rhs effectors are specifically delivered and how the encapsulated toxin CTD is released and activated. Given that we had shown that the Tse15 structure possessed an encapsulated and unfolded toxin, and dissected how Tse15 and VgrG15 interact for delivery, we were interested in what components of the Tse15:VgrG15 complex were specifically delivered out of the cell. To determine this, we re-analysed previously collected secretome data for *A. baumannii* AB307-0294 with an active or inactive T6SS (Δ*tssM*)[23]. The *A. baumannii* AB307-0294 T6SS is constitutively active, and as such, detection of the presence of T6SS effectors in the secreted fraction can be used as a proxy for delivery of effectors into prey cells. Analysis of the secretomes of the wildtype AB307-0294 strain and the Δ*tssM* mutant showed that >75 Tse15 and VgrG15 peptides were consistently identified by MS/MS across all replicates of the wild-type

secreted fraction, but none were identified in any replicate of the mutant secreted fraction (Supplementary Fig. 16). The peptide coverage for VgrG15 was across the entire protein suggesting that the entire protein is likely to be delivered into target cells. Surprisingly, we found only two very low abundance peptides from the Tse15 Rhs domain (with only one and two peptide-spectrum matches, respectively), whilst 12 and 4 unique peptides were identified from the clade and toxin domains (with a total of 85 and 14 peptide-spectrum matches, respectively). This suggests that the clade and effector domains are predominantly secreted without the Rhs domain. Absence of Rhs domain peptides is not due to protease inaccessibility as fragments across the entire Rhs domain of recombinant Tse15 were able to be identified using peptide fingerprinting. Strain AB307-0294 delivers a second Rhs effector (Tde16), therefore we also analysed the delivery of this completely different Rhs effector[23,24]. Supporting our observations for Tse15, only the clade domain (12 peptides) and effector domain peptides (42 peptides) were identified in the supernatant for Tde16, with no Tde16 Rhs domain peptides identified as secreted (Supplementary Fig. 16).

To confirm the highly reduced delivery of Tse15 (and Tde16) Rhs cage peptides, we repeated the proteomics with the addition of analysis of whole-cell lysate samples (Fig. 6). As expected, in the whole-cell lysate samples, peptides covering the majority of Tse15 and Tde16 were identified. In the supernatant samples, we identified increased numbers (0.9-1.6-fold) and intensity (2.2–7.6-fold) of peptides for the clade and toxin regions but highly reduced peptides (0.04–0.09-fold) and intensities (.07-.34-fold) for the cage peptides for both Tse15 and Tde16 (Supplementary Table 5). By comparison, for two other T6SS proteins (Hcp and Tae17) peptide coverage and intensities for peptides in the supernatants were increased across the whole proteins (1–2.3-fold for unique peptides and 4.1–6.5-fold for peptide intensities). RpoB was also analysed as a non-secreted cytoplasmic protein; whole-cell lysate samples showed coverage across the whole proteins, while very few peptides were observed in the supernatant samples, indicating the samples displayed very low levels of sample lysis. The values for RpoB in the supernatant are similar to that observed for the Tde16 Rhs cage, suggesting that the cage peptides observed for Tde16 may be primarily the result of low-level sample lysis, rather than direct secretion. For Tse15, it is likely that the cage is most commonly retained within the cell but may be secreted at very low levels. Overall, the two independent experiments show that for both Tse15 and Tde16, the clade and toxin domains are readily secreted, but the cage is most often retained within the host cell.

## Discussion

Multi-drug resistant (MDR) *A. baumannii* remains an elusive target for antibiotic treatment, and novel approaches for control of this problematic pathogen should be explored. This includes exploring alternative targets that might modulate its survival in mixed infections. As the T6SS enables some *A. baumannii* strains to outcompete bacterial competitors in some niches, understanding this system may aid in control of MDR pathogens. This work describes a detailed molecular analysis of one of the *A. baumannii* T6SS Rhs-family effectors, Tse15, elucidating both the structure and delivery mechanism. Tse15 possesses the canonical Rhs β-cocoon domain and we could map unfolded toxin density inside the interior of the β-cocoon. Additionally, we experimentally resolved the α-helical N-terminal clade domain structure that we identify in other Gram-negative bacteria that possess a T6SS. The conservation of the clade sequence and structure, as well as the cleavage motif we identify here, suggest that our data on the Tse15 structure and function may be generally applicable to homologous T6SS cargo effectors from various pathogenic and non-pathogenic bacteria.

Recent research has provided insight into the structure of Rhs cargo effectors delivered by the T6SS[26,31,32], describing a three-domain

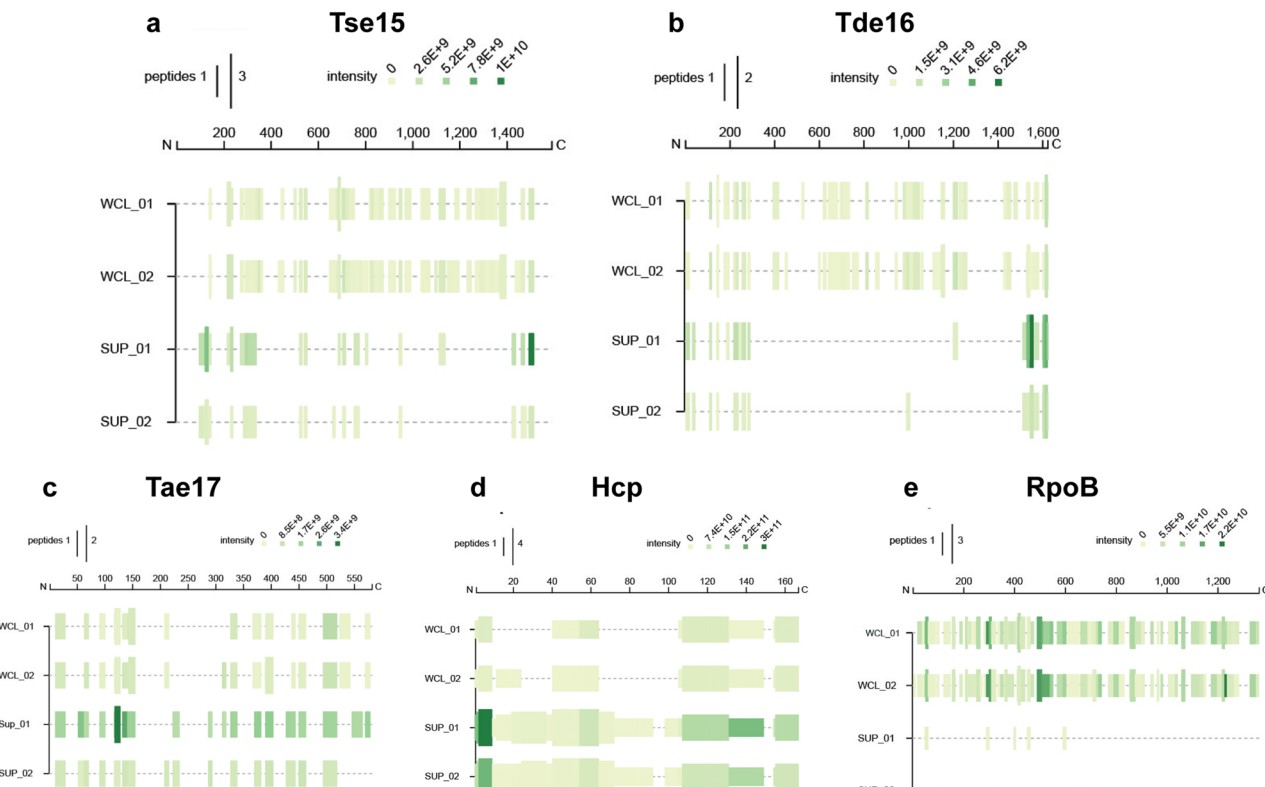

**Fig. 6 | Visualisation of secretome data from *A. baumannii* AB307-0294.** Visualisation of secretome data from *A. baumannii* AB307-0294 showing the marked absence of cage peptides for Tse15 (**a**) and Tde16 (**b**). Also shown is peptide coverage for the T6SS control proteins Tae17 (a T6SS effector that does not use a Rhs cage) (**c**) and Hcp (a T6SS structural subunit) (**d**) as well as a cytoplasmic protein RpoB (**e**). The amino acid numbers are shown at the top of each panel. Peptide coverage and intensity are indicated for each of the two replicates of whole cell lysate (WCL) and supernatants (SUP) samples. Intensity legend is in the top right of each panel.

structure. Autocleavage of the toxin domain from the Rhs cocoon via an aspartyl-protease self-cleavage event has also been reported in other Rhs- and Rhs-like proteins, such as ABC toxins[29]. We confirmed Tse15 possessed aspartyl protease-mediated autocleavage of the toxin domain but showed that this event had little effect on the overall tertiary structure of this effector. This is in contrast to Tang et al. who described an autoproteolysis-triggered conformational change that led to dimerization of RhsP from *V. parahaemolyticus*[26]. The RhsP dimer was formed as a pseudo-symmetrical anti-parallel ('head to tail') dimer via the interaction of two essential residues on the exterior of the Rhs cocoon and was essential for the T6SS prey-targeting. Günther et al. also reported the formation of dimers in RhsA from *P. protegens* but did not assign an essential mechanistic function to the species[32]. Our data strongly suggest that Tse15 is monomeric both in solution and on cryo-EM grids, regardless of toxin autocleavage. Close inspection of our grids did identify instances of two particles close together, but these particles showed highly variable shapes, suggesting that it was not an ordered dimer interaction and rather crowding of monomers within the vitreous ice layer.

During production of Rhs effector proteins, it is widely accepted that the Rhs cage is required to protect the host from self-killing. The Rhs effector is translated as one chain; and for Tse15, our data shows that the toxin remains unfolded inside the Rhs cage, lining the interior of the Rhs β-stranded sub-structures. The unfolded nature of the toxin was not completely surprising to us given the high percentage of glycine and serine residues in the toxin sequence (23.6%) that had suggested the toxin may be highly flexible. The unfolded nature of the toxin peptide was also apparent irrespective of whether the toxin had been cleaved from the main chain of the Rhs cage, strongly suggesting that the host protection is two-fold – the toxin is shielded by the Rhs

cage but also remains unfolded in a presumably catalytically inactive state prior to T6SS loading and firing. Once the T6SS is loaded and/or fired, the host no longer requires the protection of the Rhs cage as it is no longer cytosolic, or the toxin is in an inactive conformation bound to the T6SS, as was found with another Rhs effector, phospholipase Tle1[38]. Together, these events allow for the active toxin to be delivered into the target cell. For *A. baumannii* AB307-0294, the Tse15 cognate immunity protein, Tsi15, is encoded upstream of the *tse15* gene[23]. After resolution of the Tse15 structure and with the shielded, unfolded toxin resolved, we predicted that the immunity protein is not required by the host cell that is preparing to fire the T6SS but is rather produced to give immunity to injection of active Tse15 toxin from a neighbouring cell. Indeed, a Δ*tsi15*Δ*tssM* mutant unable to fire the T6SS and lacking the Tsi15 immunity protein was as viable as wild-type parent AB307-0294, showing that Tsi15 is not required to prevent self-intoxication.

What remains generally unknown in the field and indeed for Tse15 is the signal for toxin release in Rhs effectors. The trigger(s) of autoproteolysis remains unidentified. Previous studies have indicated that self-cleavage of the toxin is a signal for release into the target cell. Such a mechanism would require complex signalling to allow for synchronised toxin autocleavage and release. In AB307-0294, the effector is delivered into the supernatant constitutively. This suggests that the signal is not dependent on the presence of target cells. We propose that these events are independent. Both N-terminal clade and toxin self-cleavage occurs following Tse15 main chain protein folding as it brings the catalytic regions in proximity to their target. Following auto-cleavage, the toxin is frequently released from the Rhs cage prior to delivery into the prey cell by the T6SS. This is supported by the fact our secretome data show that the Rhs domain most often dissociates before the firing of the T6SS complex. Further evidence to show that

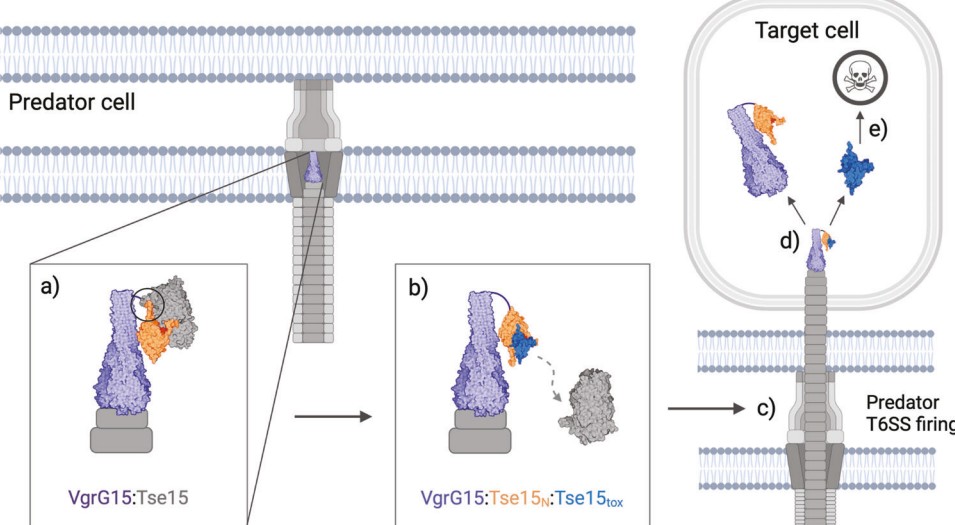

**Fig. 7 | Delivery schematic for Tse15:VgrG15.** VgrG15 is depicted as purple, Tse15 is coloured by domain where clade is orange, Rhs is grey and toxin is blue. **a** Initial interactions between the Tse15 clade and Rhs domains and VgrG15. **b** Tse15 clade domain forms edge-to-edge contact interactions with VgrG15. The clade and toxin domains of Tse15 interact, triggering the Rhs dissociation. **c** When the Tse15 clade:toxin is bound, the T6SS fires into the target cell. **d** Following delivery, the toxin dissociates from protein chaperones (clade and VgrG15). The toxin can now fold independently and illicit toxic activity to cause target cell death (**e**). Created with BioRender.com.

autocleavage of Tse15 (and likely other Rhs proteins) is not dependent on external factors, is that Tse15 autocleavage occurs even when produced in non-native *E. coli* cells. Finally, we postulate that the Rhs domain itself is rarely secreted out of the host cell due to the size of the cage as this would effectively 'blunt' the end of the T6SS.

Our T6SS competitive killing assay data clearly shows that both the clade cleavage region and the toxin cleavage region are important for Tse15 toxicity. Interestingly, the killing ability of the cleavage mutants (*tse15*CC::FRT and *tse15*TC::FRT) was not completely abolished. We hypothesize that low levels of the mutated Tse15, unable to cleave at either the clade or toxin cleavage site, are delivered into the prey cell by the T6SS system and that subsequent proteolysis of the Rhs cage allows the toxin to be activated. A similar phenomenon has been observed previously for the *Aeromonas dhakensis* T6SS Rhs effector, Tse1. Recombinant Tse1 is toxic when expressed in *E. coli* and although the protein was significantly reduced in toxicity when the clade cleavage region or toxin cleavage region was mutated it still retained some residual activity[27].

We were also able to visualise the N-terminal clade domain of Tse15. We identified the Tse15 clade domain in multiple other Gram-negative Rhs effectors and found that the cleavage motif is conserved amongst these homologues. However, the motif identified in this study diverges from those previously described in the literature. Previous analysis of Rhs-family effectors from *A. dhakensis* and *V. parahaemolyticus* identified cleavage at a proline residue with the motif x/PVxxxxGE[26,27]. In both instances, mutation of the proximal glutamic acid residues prevented N-terminal autocleavage, suggesting that this residue is crucial for the autocleavage event[26,27]. The *A. baumannii* motif we determined consists of 'x/(S/N)Ix4G(T/A)Ex3HxD', with a glutamic acid also +7 positions from the cleavage site. Using mutagenesis, we demonstrated that a proximal glutamic acid is also important for autoproteolysis of the Tse15 clade domain. Mutation of the P1-P1' cleavage site residues, Tse15 K334 and S335, reduced cleavage efficiency but did not stop it altogether, demonstrating that conservation of the cleavage site is not critical for autocleavage. As conservation of the glutamic acid was crucial, we propose this residue acts as the nucleophile for cleavage of the peptide bond.

Some Rhs effectors utilise a chaperone in order to permit effector delivery[31,32]. However, there is no known chaperone associated with Tse15, and no chaperone-encoding gene has been identified in the *tse15* locus. Our data suggest that the N-terminal clade domain is required for delivery and appears to act as an internal chaperone once released from the body of the molecule by autocleavage. The Tse15 N-terminal domain appears to interact with a VgrG15 Ig-like domain using edge-to-edge strand contacts. A very similar interaction has been experimentally confirmed in the an enteroaggregative *E. coli* VgrG and its cognate Tle1 phospholipase (non-Rhs) cargo effector protein[38]. Both this VgrG domain, described as a transthyretin like (TTR) domain, and our VgrG15 Ig-like domain, form a β-strand rich sandwich structure, that forms a three stranded anti-parallel β-sheet on one side. One strand of this sheet forms the edge-to-edge contacts between the VgrG protein and the effector, where the effector forms a smaller β-strand to mediate these interactions. The strong similarity between the VgrG15 Ig-like domain and the *E. coli* VgrG 'TTR' domain, even though Tse15 and Tle are structurally distinct effectors, suggests that VgrG cargo effector binding domains may be highly conserved regardless of effector structure/function.

Taken together, our data allow us to present a clearly defined mechanism for Rhs effector delivery. The Rhs cage is first responsible for delivering the toxin to VgrG, with Tse15:VgrG15 interactions involving both the N-terminal clade and Rhs domains of Tse15 (Fig. 7a). Once Tse15 is bound to VgrG15, we postulate that the Rhs domain dissociates immediately and is retained inside the cell when the system fires or infrequently can still be fired from the cell (Fig. 7b). In both scenarios, a large conformational shift must occur either before or during firing. This conformational shift may be mediated by changes to salt bridges across the N-terminal clade and Rhs domain interface. Importantly, interactions between the clade domain and the toxin CTD within the Rhs cage allow these domains to remain associated if the Rhs domain dissociates. Our structure shows a gap between two of the Rhs β-cocoon substructures in proximity to the N-terminal clade domain. This region may allow for the toxin to interact with the N-terminal domain, and subsequently exit the Rhs cage. During this complex rearrangement and priming of the T6SS by VgrG effector loading, conformational changes would be required and account for the flexibility observed within models of clade and VgrG15 (Fig. 7b, c). Our AlphaFold2 model also shows an edge-to-edge contact interaction between a predicted Ig-like domain of VgrG15 and the clade domain of

Tse15; however, our VgrG deletion experiments show that this alone is not sufficient for Tse15 delivery. If the modelled edge-to-edge contact exists, this interaction strength, together with further interactions with the C-terminal region of VgrG15, makes it feasible that VgrG15, the Tse15 clade domain and bound toxin remain associated during delivery into the target cell (Fig. 7c, d). Separation of the clade domain from the toxin CTD likely permits final effector folding, independently of the T6SS and associated proteins, allowing the effector to illicit its toxic activity (Fig. 7e).

Overall, this study has presented a T6SS Rhs effector structure that showed an unfolded toxin within the Rhs domain β -cocoon and determined the structure of an α-helical N-terminal clade domain. We also mapped the interactions of Tse15 with its cognate VgrG15, and our secretome data show that the Rhs cage of both AB307-0294 Rhs proteins (Tse15 and Tde16) are often not secreted outside of the predator cell. Thus, we propose that the Rhs domain of Tse15 is essential for delivering the effector to VgrG15 and initiating the binding interaction. However, the Rhs domain most often then dissociates or stays associated until firing, upon which physical interactions with the predator cell membrane and/or T6SS membrane structures forces the Rhs domain to dissociate. These events allow the N-terminal clade domain together with the CTD toxin domain to exit the predator cell. This model is in contrast to previous hypotheses that entire Rhs effectors are always delivered into the prey cell[32]. As the interactions between VgrG and the effector/effector chaperone may be conserved between cargo effectors, we propose that this is a general mechanism for delivery of Rhs effectors. However, it is still unclear what large-scale conformational changes are required to release the Rhs domain and also what specific interactions allow the clade domain to strongly interact with the CTD toxin such that they stay together during delivery into the target cell. Our deletions analyses, bacterial two-hybrid data and AlphaFold2 models, suggest that Tse15 interactions with both a VgrG15 Ig-like domain and also N-terminal α-helices are crucial for this interaction. Finally, the fact that the Rhs cage is commonly left behind fundamentally changes our understanding of toxin release and how future proteins may be engineered for delivery of bespoke T6SS payloads. The modular nature of Rhs proteins, like Tse15, may allow fusion or substitution of the C-terminal toxin domain, allowing delivery of chosen molecules to manipulate the bacteria or bacterial niche, which may aid in control of MDR pathogens.

## Methods

### Molecular biology and reagents

All plasmids and strains are listed in Supplementary Table 6. Primers used throughout this study are listed in Supplementary Table 7.

### Tse15 construct design

The full-length Tse15 (ABBFA_02439: GenBank ATY44872.1: Uniprot A0A5K6CSR3) expression construct was synthesised to encode wild-type Tse15 with an in-frame N-terminal StrepTag-II and linker as well an in-frame C-terminal hexa-histidine tag (MA-WSHPQFEK-SA-Tse15-HHHHHH). The gene sequence was codon optimised for expression in *E. coli* and synthesised by GenScript. The gene was then cloned by GenScript into a pET-28a+ vector using a 5′ NcoI and 3′ BamHI restriction sites. Variants of Tse15 were generated by GenScript via site-directed mutagenesis.

### Construction of *A. baumannii* mutants

PCR-mediated recombination and mutagenesis of *A. baumannii* strain AB307-0294 was performed by introducing splice-overlap PCR (SOE-PCR) products into electrocompetent cells. SOE-PCR products encoding the desired change and a central kanamycin cassette flanked by FRT sites (to allow for removal of the kanamycin cassette) were used for mutagenesis as described previously[39]. For deletion of *tsi15*, SOE-PCR products lacking FRT sites were used for mutagenesis as described previously[40]. Primers used for the construction of each mutant are listed in Table S7

### Construction of plasmids used in T6SS competitive killing assays

The gene encoding the chimera VgrG16_15$_{831-1064}$ was constructed using high-fidelity SOE-PCR. Primers BAP8527 and BAP8582 were used for PCR with template pAL1416[23], to amplify *vgrG16* nucleotides 1498-2491. Primers BAP7767 and BAP8581 were used to amplify the 3′ region of *vgrG15* (from nucleotide 2441) with pAL1415 as template. The first round PCR products were then used as templates in a second round PCR with flanking primers BAP8527 and BAP7767 to produce the C-terminal region of the chimera. The SOE-PCR product was digested with EcoRI and ligated to similarly digested pAL1458 encoding *vgrG16* nucleotides 1-1597 (up to a native internal EcoRI site). Each ligation was used to transform competent *E. coli* DH5α cells and transformants were selected with 100 μg/mL ampicillin. The correct plasmid, pAL1456, was identified using Sanger sequencing. To determine if the chimeric protein was able to facilitate Tse15-mediated killing of *E. coli* in T6SS killing assays, the plasmid was introduced into the *A. baumannii vgrG15* mutant AL2751.

Plasmids, pAL1471, pAL1473 and pAL1474, each encoding a truncated VgrG15 protein were constructed by ligating EcoRI-digested PCR products representing regions of the C-terminal region of VgrG15 (from amino acid 531 onwards) to a VgrG15 EcoRI subclone, pAL1457, encoding only the N-terminal region of VgrG15 (amino acids 1 to 530). As an example, the plasmid pAL1471 (Table S6) encoding VgrG15$_{1-912}$ was constructed by ligating EcoRI-digested PCR product encoding VgrG15 amino acids 531-912 followed by a stop codon (generated with pAL1415 template and primers BAP8527 and BAP8606, Table S7) to EcoRI-digested pAL1457. Each ligation was used to transform competent *E. coli* DH5α cells and transformants were selected on LB agar with 100 μg/mL ampicillin. Each plasmid was confirmed to contain the correct insert using Sanger sequencing with vector primers flanking the cloning site (UP and BAP8412) and *vgrG15*-specific primers (BAP8527 and BAP8528). To determine if each truncated VgrG15 protein was able to facilitate Tse15-mediated killing of *E. coli* in T6SS competitive killing assays, each plasmid was separately introduced into the *A. baumannii* ΔvgrG15 mutant AL2751.

The expression plasmid pAL1707 encoding full length VgrG15 with multiple alanine substitutions (L999A, N1000A, L1003A, S1004A, Q1007A, M1008A, L1011A) was constructed by replacing (via restriction enzyme digestion and ligation), the relevant wildtype sequence in pAL1415 with a high-fidelity PCR product encoding all desired codon changes (generated using BAP9144 and BAP9199). Sanger sequencing confirmed the expected mutations were present in pAL1707. For competitive killing assays, the plasmid was used to transform electrocompetent *A. baumannii* ΔvgrG15 mutant, AL2751.

### Construction of bacterial two-hybrid plasmids

PCR products containing sections of *vgrG15* or *tse15* were digested with PstI and XmaI and cloned into similarly digested pUT18C or pKT25 (Table S6), such that the adenylate cyclase T18 and T25 open reading frames (encoded on pUT18C and pKT25, respectively) were upstream and in-frame with the open-reading frames in the cloned fragments. For example, pAL1551 was generated by ligating PstI/XmaI-digested pKT25 to a similarly digested PCR product encoding Tse15 amino acids 2-1389, generated using primers BAP8800 and BAP8801 (Table S7) with AB307-0294 genomic DNA as template. Each ligation was used to transform competent *E. coli* DH5α cells and transformants were selected on LB agar supplemented with 100 μg/mL ampicillin (pUT18 ligations) or LB agar supplemented with 50 μg/mL kanamycin (pKT25 ligations). Transformants containing the correct recombinant plasmids were confirmed by Sanger sequencing using vector-specific primers flanking the insert.

## Construction of multi-alanine substituted VgrG15

Two-hybrid plasmids for alanine scanning of VgrG15$_{976-1024}$ were constructed using PCR products generated with a series of extended primers covering *vgrG15* nucleotides 2926–3072. To introduce single or double alanine substitutions, the forward or reverse primer was modified to include a single codon change, or a consecutive codon change to GCn. Where the natural codon encoded alanine, the codon was changed to encode glycine. Each PCR product encoding the substitution/s was cloned into PstI/XmaI-digested pUT18C as described above. The authenticity of the cloned fragments was confirmed by Sanger sequencing.

## Proteomic analysis of *A. baumannii* strain AB307-0294

Duplicate late-log phase cultures of *A. baumannii* strain AB3037-0294 were prepared as follows. Each 100 mL volume of M9 media, supplemented with 1% casamino acids (Bacto, Australia) and 20 mM sodium butyrate (Merck KGaA, Germany), were inoculated with 200 μL of stationary-phase AB3037-0294 cells (washed in 1 x volume M9 media) and the culture grown at 37 °C with shaking (200 rpm) to an optical density (600 nm) value of 0.8. Concentrated culture supernatant samples were prepared for secreted protein analysis as described previously[23]. Briefly, 100 mL of each late-log phase culture (*n* = 2) was centrifuged (10 min, 4000 x g at 4 °C) and the supernatant collected and filtered through a 0.22 μM vacuum filter (Corning, USA) to remove cellular material. The supernatant was then concentrated (~200 x) using a Amicon Ultra-15 Centrifugal Filter (3 kDa molecular weight cutoff, Merck KGaA, Germany).

For cell analysis, 1 mL of each culture was centrifuged (10,000 *rcf*, 1 min), and then cells were washed with 1 mL of PBS before being collected by centrifugation. Washed cell pellets and concentrated supernatant samples were stored at −80 °C until analysis at the Monash University Proteomics and Metabolomics Platform. Analysis was performed on two biological samples (*N* = 2) for both the whole-cell proteome and the respective secretome. The Monash Proteomics and Metabolomics Platform prepared all samples under standard procedures described here. Secretome samples were centrifuged at 4000 *rcf* for 5 min to remove insoluble material, followed by filtration and concentration using Amicon Ultra Centrifugal Filter Devices 15 kDa (Merck, Millipore). Whole-cell proteome samples were solubilized in SDS lysis buffer (5% w/v sodium dodecyl sulfate, 100 mM HEPES, pH 8.1), heated at 95 °C for 10 min, and sonicated to shear the DNA in three 30 s intervals with resting periods on wet ice. Samples were clarified by centrifugation before determining total protein content using a BCA kit (Thermo Fisher) according to the manufacturer's instructions. Normalised amounts of protein were denatured and alkylated by adding TCEP (Tris(2-carboxyethyl) phosphine hydrochloride) and CAA (2-chloroacetamide) to a final concentration of 10 mM and 40 mM, respectively, and the mixture was incubated at 55 °C for 15 min. Samples were processed via the S-trap protocol according to the manufacturer's instructions[41]. Sequencing grade trypsin (Promega, Gold) was added at an enzyme-to-protein ratio of 1:50 and incubated overnight at 37 °C after the proteins were trapped using S-Trap mini columns (Profiti). The peptide samples were cleaned up by STAGE-tips packed with SDB-RPS (Empore)[42]. All samples were acidified with 0.1% formic acid upon reconstitution and spiked with iRT peptides.

Peptide analysis was performed utilizing a Dionex UltiMate 3000 RSLCnano system equipped with a Dionex UltiMate 3000 RS autosampler, an Acclaim PepMap RSLC analytical column (75 μm x 50 cm, nanoViper, C18, 2 μm, 100 Å; Thermo Scientific) and an Acclaim PepMap 100 trap column (100 μm x 2 cm, nanoViper, C18, 5 μm, 100 Å; Thermo Scientific), with tryptic peptides separated by increasing concentrations of 80% acetonitrile (ACN) / 0.1% formic acid at a flow of 250 nL/min for 120 min of linear separation and analysed with a Orbitrap Exploris 480 mass spectrometer equipped with a FAIMS module (Thermo Scientific). The instrument was operated in data-dependent

acquisition mode to automatically switch between full scan MS and MS/MS acquisition using two sequential compensation voltages of -45 V and -75 V. Each survey full scan (m/z 350–1200) was acquired in the Orbitrap with a resolution of 60,000 (at m/z 200) after accumulating ions with a normalized AGC (automatic gain control) target of 300% and an automated maximum injection time. The most intense multiply charged ions (z ≥ 2) within 1.7 sec and 1.3 sec (for compensation voltages of -45 V and -75 V, respectively) were sequentially isolated and fragmented in the collision cell by higher-energy collisional dissociation (HCD). All ms2 scans were acquired with a resolution of 15,000, a normalized AGC target of 75 %, using an automated maximum injection time. Dynamic exclusion was set to 45 s and shared across compensation voltages.

The protein identification and quantification were performed by using Fragpipe with MSfragger[43] as the search engine against the *A. baumannii*, strain AB307-0294, database from SwissProt (March 2024). The standard label-free quantification match-between-runs (LFQ-MBR) workflow was applied with no changes to workflow, employing IonQuant[44] and the MaxLFQ method of protein abundance determination. Percolator was used for PSM validation, with strict Trypsin as the enzyme allowing up to two missed cleavages. A 1% FDR cutoff was applied at the protein level, peptide level and matching-between-runs identification transfer. Statistical analysis and visualisation were performed using standard settings in LFQ-Analyst from the Monash Proteomic Analyst Suites (https://analyst-suites.org/)[45]. The mass spectrometry proteomics data have been deposited to the ProteomeXchange Consortium via the PRIDE partner repository with the dataset identifier PXD05195.

## T6SS competitive killing assays

T6SS competitive killing assays used to investigate VgrG15 chimera, truncated proteins, and alanine substituted VgrG15, were performed as described previously[23] with the exception that the assay incubation time was 3 h. To specifically detect Tse15-mediated killing, indicative of delivery of a functional VgrG15, the other two T6SS effectors of strain AB307-0294, Tde16 and Tae17, were neutralised by expression of their cognate immunity protein in the *E. coli* prey cell. This was achieved by using an *E. coli* prey strain harbouring pAL1265, a plasmid encoding immunity proteins Tdi16 and Tai17 as described previously[23]. This strain is fully viable in the presence of the *A. baumannii* Δ*vgr15* mutant AL2751. To determine the ability of modified VgrG15 to facilitate delivery of Tse15 into *E. coli* prey cells, predator strains were generated as follows. Plasmids encoding each modified VgrG15 protein were separately used to transform electrocompetent *A. baumannii* Δ*vgr15* mutant AL2751. As controls, vector only and wildtype VgrG15 (pAL1415) were also separately used to transform AL2751. For assessing the Tse15 clade cleavage and toxin cleavage mutants, each mutant was used separately against *E. coli* prey strain harbouring pAL1265. As controls, wild-type AB307-0294, the *tssM* mutant (unable to produce a T6SS), and the *tse15* deletion mutant (unable to produce the Tse15 effector) were used as predator strains. All *A. baumannii* strains were tested a minimum of four times. One-way ANOVA statistical analysis was performed followed by Tukey's multiple comparisons test using $\log_{10}$ transformed values of surviving *E. coli* CFUs following the T6SS killing assay. A *P*-value of <0.05 was accepted as statistically significant. For all assays, CFU values for each *A. baumannii* predator strain grown in the presence of *E. coli* were compared to values when grown in media alone. No loss of *A. baumannii* predator strain viability was detected in any of the assays performed.

## Bacterial adenylate cyclase two-hybrid assay

The bacterial adenylate cyclase assay was performed as described previously[46] with the following modifications. BTH101 *E. coli* cells co-transformed with pUT18C and pKT25 plasmid derivatives were initially recovered on LB agar with 100 μg/mL ampicillin and 50 μg/mL kanamycin. A minimum of three separate colonies from each

transformation were used to inoculate separate LB broths supplemented with 100 µg/mL ampicillin, 50 µg/mL kanamycin and 0.5 mM Isopropyl ß-D-1-thiogalactopyranoside (IPTG). Cultures were grown overnight at 30 °C with shaking. For each overnight culture, a 10 µL aliquot

was spotted onto LB agar supplemented with 100 µg/mL ampicillin, 50 µg/mL kanamycin, 0.5 mM IPTG and 40 µg/mL bromo-chloro-indolyl-galactopyranoside (Xgal) and incubated at 30 °C for 1–2 days. Blue coloured growth in three or more replicates was indicative of interaction between the T18 and T25 fusion proteins. For an interaction to be scored as positive, the entire colony displayed a blue/green colour. No interaction was scored when the colonies on the plate retained the typical cream colour (generally with a blue/green edge). An example of positive, negative and intermediate plates are provided in Fig. S14.

### Recombinant protein purification

Expression constructs were transformed and subsequently expressed in Rosetta 2 *E. coli* cells (Novagen). Expression was conducted using 2YT autoinduction media under kanamycin selection at 30 °C for 16–20 h. *E. coli* cells were lysed by sonication (5 × 30 s, amplitude 10%) in lysis buffer (1 x PBS pH 7.4, 0.3 M NaCl, 5% glycerol, 20 mM imidazole). The resulting homogenate was clarified by centrifugation at 20,000 x *g* for 30 min at 4 °C. The soluble material was subjected to nickel affinity using a 5 mL HisTrap™ HP column (Cytiva) equilibrated with lysis buffer. The bound protein was washed using lysis buffer and eluted using lysis buffer with 250 mM imidazole. Purified protein was applied to a gel filtration step using a Superdex200 10/300 GL Increase column (Cytiva) equilibrated with 50 mM Hepes pH 8.0 and 0.3 M NaCl using an AKTA GO chromatography system (Cytiva). Purified protein was frozen in gel filtration buffer in aliquots and stored in at -80 °C until use.

### Cross-linking mass spectrometry

The analysis was conducted on two samples, one control (without crosslinker) and one experimental (with crosslinker), each with a single biological replicate ($n = 1$). For the experimental sample, Tse15 was diluted to 5 µM in 50 mM Hepes pH 8.0 and 0.3 M NaCl. BS3 crosslinker (bis(sulfosuccinimidyl)suberate; Thermofisher) was added to a concentration of 0.5 mM (1:100 protein:crosslinker). This was incubated at room temperature for 20 min before Tris pH 8 was added to a final concentration of 50 mM to quench the reaction. This was incubated at room temperature for a further 30 min before being snap frozen prior to analysis. After thawing, samples were denatured by the addition of DTT (final concentration of 10 mM) and incubation at 65 °C for 30 min. Samples were alkylated by addition of chloroacetamide (CAA) at 40 mM followed by a 30 min incubation at room temperature in the dark. Proteins were digested using trypsin (1:100 w/w trypsin:protein) (Promega, Gold) at 37 °C overnight. Digestion was halted by the addition of 1% formic acid (v/v), and peptides cleaned up by solid phase extraction as described above prior to mass spectrometric acquisition. The control sample performed the same procedure without adding BS3.

The samples were analysed by the same liquid chromatography as previously mentioned with a gradient of 90 min of linear separation and paired with an Orbitrap Fusion Tribrid instrument (Thermo Scientific). The mass spectrometer was operated in data-dependent acquisition mode keeping the cycle time controlled for 2 s. The MS1 resolution was set at 120,000 and a scan range of 375 – 2000 m/z. The normalised AGC target was set to 250% with a maximum injection time of 118 ms. The MS2 resolution was set at 60,000 with a normalised AGC target of 800% and a dynamic injection time. pLink2 (v2.3.9) was used identify BS3-crosslinked peptides species[47]. Resulting cross linked peptides were analysed and included if the peptide occurred more than once with a *P*-value $< 10^{-4}$.

### N-terminal sequencing and protein identification

For N-terminal sequencing, ~ 2.5 µg of protein was resolved on an SDS-PAGE gel and transblotted onto SequiBlot PVDF membrane (BioRad) in 10 mM CAPS pH 11 buffer. The membrane was stained with 0.025% Coomassie Blue R250 in 40% v/v methanol, and excess stain was removed with 40% v/v methanol alone. Bands of interest were excised and the sequence of the first 8 amino acids analysed by Edman degradation using a PPSQ-53a Protein Sequencer according to manufacturer's instructions (Shimadzu). Protein identification was conducted using bands excised from an SDS-PAGE gel stained with Coomassie Blue R250. Gel bands were manually excised and de-stained with a solution of 50 mM ammonium bicarbonate and 50% acetonitrile. The protein was reduced in 2.5 mM DTT at 90 °C for 15 min followed by alkylation with 10 mM CAA for 30 min at room temperature. The gel pieces were washed and dehydrated with alternating washing cycles of 50 mM ammonium bicarbonate and acetonitrile. After complete dehydration of the gel piece, it was rehydrated with a solution containing 0.5 mg trypsin (Promega corp., Madison, WI, USA) in 20 mM ammonium bicarbonate. The gels pieces are incubated at 37 °C overnight and sonicated for 2 min prior to analysis by LC-MS.

Tryptic digests were analysed by LC-MS/MS using the QExactive HF mass spectrometer (Thermo Scientific, Bremen, Germany) coupled online with an Ultimate 3000 RSLCnano system (Thermo Scientific, Bremen, Germany). Samples were concentrated on an Acclaim Pep-Map 100 (100 µm × 2 cm, nanoViper, C18, 5 µm, 100 Å; Thermo Scientific) trap column and separated on an Acclaim PepMap RSLC (75 µm × 50 cm, nanoViper, C18, 2 µm, 100 Å; Thermo Scientific) analytical column by increasing concentrations of 80% acetonitrile/0.1% formic acid at a flow of 250 nL/min for 90 min. The mass spectrometer was operated in the data-dependent acquisition mode to automatically switch between full scan MS and MS/MS acquisition. Each survey full scan (m/z 375–1575) was acquired in the Orbitrap with 60,000 resolution (at m/z 200) after accumulation of ions to a $3 \times 10^6$ target value with maximum injection time of 54 ms. Dynamic exclusion was set to 15 s and the 12 most intense multiply charged ions ($z \geq 2$) were sequentially isolated and fragmented in the collision cell by higher-energy collisional dissociation (HCD) with a fixed injection time of 54 ms, 30,000 resolution and automatic gain control (AGC) target of $2 \times 10^5$.

The raw files were analysed using the Byonic v3.0.0 (Protein-Metrics) search engine and searched against a custom database of the recombinant sequence appended to the *E. coli* strain B / BL21-DE3 from UniProtKB to obtain sequence information. Only proteins falling within a predefined false discovery rate (FDR) of 1% based on a decoy database were considered further. Semi specific cleavage sites were specified and a precursor and fragment mass tolerance of 20 ppm. Modifications specified were Carbamidomethyl @C fixed and Oxidation @M Variable common 1.

### Pulldown of native VgrG15 using recombinant Tse15

For the native expression of VgrG15, 2 L of AL4693 *A. baumannii* AB307-0294 cells harbouring pAL1415 (Table S6) were grown in 2 L flasks (400 mL per flask) for 7 h at 37 °C with 200 rpm shaking. The cells were collected by centrifugation at 5000 x *g* for 10 min at 4 °C and stored at -80 °C until use. For the pulldown, 200 µL of loose nickel resin (Qiagen) was pre-equilibrated with wash buffer (50 mM Hepes pH 8, 0.15 M NaCl, 20 mM imidazole), then 0.4 mg of purified Tse15 was incubated with the resin for 1 h at 4 °C with gentle agitation. Following this, the resin was washed with 20 CV of wash buffer. For the pulldown of VgrG15 from *A. baumannii*, the cells were resuspended in 80 mL of lysis buffer (wash buffer and protease inhibitor tablet (Roche) then sonicated on ice with an amplitude of 10, for 30 sec on, 1 min off for 6 cycles. Clarified lysate was obtained by centrifugation at 38,000 x *g* at 4 °C for 30 mins. The clarified lysate was added to the nickel resin with

Tse15 bound using gravity flow. The resin was washed with 20 CV of wash buffer, and then eluted in 1 mL with wash buffer with 250 mM imidazole. The resulting elution was concentrated to 100 μL using a 30,000 MWCO concentrator (Merck) and boiled with SDS-loading dye before analysis using SDS-PAGE. Following SDS-PAGE, all bands were excised and protein identity was assessed using mass spectrometry-based peptide fingerprinting.

## AlphaFold2 modelling

AlphaFold2 models were produced using AlphaFold2 version 2.1.1 on the MASSIVE M3 computing cluster[48]. For Tse15, models were produced in monomer mode and the full-length sequence used for input. For Tse15$_{NN}$, models were produced in multimer mode where residues 1–334 comprised one chain, and residues 335–1590 were used as the second chain. Five models were produced, with an unrelaxed and relaxed output for each. For the monomer mode, the relaxed models were ranked on their pLDDT scores, and the best pLDDT score was used for model building. For the multimer mode, relaxed models were ranked using an iPTM (or DockQ) score. The top ranked model (by iPTM score) was used for model building.

For VgrG15:Tse15 multimer modelling, VgrG15 residues 851–1064 were used as one chain and full length Tse15 as a second chain (iPTM 0.74). To create the model of the VgrG15 trimer and Tse15 interacting, the N-terminal 19 - 585 residues of VgrG15, which are homologous to solved VgrG structures, were modelled as a trimer using SWISS-MODEL homology modelling techniques (reference structure: 6H3N, 25.5% sequence identity) (references on SwissM website). The C-terminal 578–1064 residues were then modelled by AlphaFold2 multimer to also produce a trimer where the end 107 residues were trimmed due to disorder (iPTM 0.69). The N-terminal and C-terminal VgrG15 homotrimer components were then overlayed using Chimera. To attach Tse15, the multimer model of VgrG15:Tse15 interacting was then superimposed onto one C-terminal chain of the VgrG15 trimer. The structure of Tse15 wildtype was then superimposed with the Tse15 N-terminal region modelled by the VgrG15:Tse15 interaction.

## Cryo-electron microscopy

For grid preparation, 3 μL of 3 mg/mL of protein was added to glow discharged UltrAuFoil holey gold grid (Quantifoil GmbH) using a Vitrobot Mark IV (Thermofisher) set at 4 °C and 100% humidity. Excess sample was blotted for 3 s using filter paper (blot force -2) and grids were flash frozen in liquid ethane. For Tse15, data were collected from grids using a FEI Talos Artica (Thermofisher) operated at an accelerating voltage of 200 kV. Images were acquired at a 50 μm C2 aperture at a nominal magnification of 150kx resulting in a pixel size of 0.94 Å/pixel on a Falcon 3EC direct electron detector (Thermofisher). A total of 2205 movies each with an accumulated dose of 50 e/Å² were collected fractionated into 50 frames. EPU (Thermofisscher) was used for automatic data collection using beam image shift data collection to image 9 holes per stage move. For Tse15$_{NN}$, data was collected using a G1-Titan Krios (Thermofisher) operated at 300 kV equipped with Gatan K3 mounted post Gatan Bioquantumn energy filter. Images were collected at a magnification of 130kx EFTEM mode resulting in a pixel size of 0.65 Å/pixel. Zero loss filtering was done using a narrow-slit width of 10 eV. A total of 5049 movies were collected using a K3 Direct Electron Detector (Gatan) operated in CDS mode. Each movie had a total dose of 60 e/Å² which were further fractionated into 60 frames. EPU was used for automatic data collection using aberration free image shift. After data collection, movies from all datasets were motion corrected using UCSF MotionCorr 2.1.4.2[49] and CTF was estimated with CTFfind-4.1.14[50] through Relion 3.1.2[51] wrappers.

## Tse15 reconstruction

Particles were picked using Gautomatch V 0.53 (developed by K. Zhang) with a diameter of 100 Å. The resultant coordinates were imported to Relion 3.1.2[51] and was extracted unbinned. The particles were then imported to cryoSPARC v3.0.1[52]. One round of 2D classification was performed to weed out junk particles. Retained particles were then subjected to ab initio reconstruction to generate an initial model. Further homogenous refinement was performed to get 3.31 Å reconstruction. The refined coordinates were imported back to Relion 3.1.2 using UCSF pyem code[53] followed by Bayesian polishing. Polished particles were imported back to cryoSPARC v3.0.1 for further processing. Heterogenous refinement was performed resulting in two distinct classes, one with and the other without the clade domain. These particles were treated separately as described in workflow (Supplementary Fig. 2). Multiple rounds of heterogenous refinement were performed to retain good quality particles. These were further refined using non-uniform refinement followed by CTF refinement and final round of non-uniform refinement to yield 3.08 Å reconstruction for Tse15 with the clade domain and 2.85 Å reconstruction for Tse15 missing the clade domain (FSC = 0.143, gold standard).

## Tse15$_{NN}$ reconstruction

Particles were picked as described for Tse15 and were extracted in Relion 3.1.2 with 4 times binning. Resultant particles were imported to cryoSPARC 3.3.2[52] and were subjected to 2D classification. A subset of particles from selected good classes were subjected to another round of 2D classification followed by 3D ab initio classification as shown in workflow (Supplementary Fig. 10). Classes 1 and 3 were used as initial model to perform heterogenous refinement on particles from selected classes from the full dataset. This resultant class showing clear secondary structural features were then refined using homogenous refinement and the particles were re-extracted in Relion 3.1.2 based on the refined coordinates with 2 times binning. These particles were then re imported to cryoSPARC 3.3.2 and were subjected to heterogenous refinement with initial models corresponding to Class 1,3 and 4 from ab initio classification. Class 1 with clade domain which showed clear secondary structural elements were subjected to homogenous refinement which resulted in a map restricted by Nyquist limitation imposed by binning. The resultant particles were then re-extracted unbinned and refined to yield a 1.89 Å map. The FSC showed signs of minor particle duplication. The particles were then re imported back to relion 3.1.2 to perform Bayesian polishing. The resulted polished particles were reimported back to cryoSPARC 3.3.2 and were subjected to multiple rounds of heterogenous refinement followed by non-uniform refinement, CTF refinement and final non-uniform refinement to yield 1.77 Å map (FSC = 0.143, gold standard).

Maps were inspected and chirality was adjusted using UCSF Chimera[36]. Model building proceeded using the top AlphaFold2 model that was fitted to the cryoEM map density using the fitmap function of UCSF Chimera. Model refinement was conducted using PHENIX real space refinement (version 1.19.2)[54] guiding manual model adjustments and de novo peptide placement that were performed using Coot 0.8.9.1[55]. Model quality was assessed using Molprobity[56] and model images were produced using Pymol 2.3.2, UCSF Chimera 1.15 and ChimeraX v1.6.1.

## Reporting summary

Further information on research design is available in the Nature Portfolio Reporting Summary linked to this article.

# Data availability

The data that support this study are available from the corresponding authors upon request. The cryo-EM maps have been deposited in the Electron Microscopy Databank (EMBD) under accession codes EMD-42792 (Tse15) and EMD-42775 (Tse15$_{NN}$). The atomic coordinates for Tse15 and Tse15$_{NN}$ are available from the Protein Data Bank (PDB) under accession codes 8UY4 (Tse15) and 8XUT (Tse15$_{NN}$). For structural comparison of Tse15, the following PDB codes were utilised:

7PQ5, 7Q97, 8H8A, 8H8B, 4IGL and 6FB3. The mass spectrometry proteomics data have been deposited to the ProteomeXchange Consortium via the PRIDE partner repository with the dataset identifier PXD05195.

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

## Acknowledgements

This work was supported by NHMRC grants 1165036 (to JDB and MH) and 1128981 (JDB, SM and MH). BKH is supported by RTP scholarship from Australian Department of Education. We thank Jordan Thompson for construction of the *A. baumannii* mutant AL4020 and Laura D'Andrea for experimental assistance with crosslinking mass spectrometry; Daniel Williams for assistance with AlphaFold2 and Clara Bate for preparation of Fig. 7. This study used the MASSIVE M3 server and BPA-enabled (Bioplatforms Australia) / NCRIS-enabled (National Collaborative Research Infrastructure Strategy) infrastructure located at the Monash Proteomics and Metabolomics Platform. We acknowledge the use of instruments and assistance at the Monash Ramaciotti Centre for Cryo-Electron Microscopy, a Node of Microscopy Australia. ARC LIEF grants (LE200100045, LE120100090) for the Titan Krios Gatan K3 Camera and for the Titan Krios.

## Author contributions

B.K.H., M.H., J.D.B. and S.M. conceived the project and designed the experiments. B.K.H conducted all recombinant protein experiments. H.V. conducted cryo-EM data collection and analysis. B.K.H, H.V. and S.M. conducted the structural modelling. M.H., J.D.B. J.M.L., A.W. conducted molecular microbiology experiments; H.L., J.R.S., D.L.S. and R.B.S. conducted proteomic analysis. All authors contributed to relevant data analysis. B.K.H, M.H., J.D.B. and S.M. wrote the manuscript. J.D.B. and S.M. supervised and funded the project.

## Competing interests

The authors declare no competing interests
