## [Peer Review File · Nature Communications]

Structure of a Rhs effector clade domain provides mechanistic insights into type VI secretion system toxin deliveryREVIEWER COMMENTS

Reviewer #1 (Remarks to the Author):

In this manuscript, Hayes et al. report cryoEM structure and autoprocessing of Tse15, a T6SS-related toxin from *Acinetobacter baumannii*. The authors show that Tse15 is composed of three domains, one being the Rhs cocoon structure holding a C-terminal toxin domain in an inactive form. Autoprocessing likely releases and activates the toxin. The authors also solved a structure of a mutant where the autoprocessing was blocked by mutation. This provided further insights into packaging of the toxin into the Rhs cocoon. Interestingly, the authors also identified another conserved autocleavage sequence that processes the N-terminal domain independently of the C-terminal processing of the toxin. In addition, the authors identified how Tse15 interacts with its cognate VgrG using a bacterial two-hybrid system. Furthermore, this interaction was suggested to be required for prey killing by the T6SS. Finally, MS analysis suggested that the Rhs cocoon is not secreted out of the cells while both the toxin and the N-terminal domain are.

The manuscript is well written and data are clear. I also think that this describes an interesting mechanism and advances our understanding of secretion and activation of T6SS-related Rhs toxins. While I cannot judge all the technical details of the structural analysis, this seems very reasonable. I would however say that primary data, quantification and statistical analysis should be included for all killing assays mentioned in the text (CFU counts). Similarly, for the two-hybrid assays, some examples of how the reporter strain looks on a plate with interacting proteins and with proteins that fail to interact (include positive and negative controls). The authors should provide detailed criteria or examples to illustrate how they differentiated between positive and negative results in their assays. Finally, the last section of the paper should be renamed. The presented data (reanalysis of previous data) support the model that the cocoon is not secreted but say nothing about target cells. The authors propose that the Tsi15 immunity protein is required only for sister cell killing. This could be tested as the immunity protein should be possible to delete (in the presence of active Tse15) in T6SS negative strain (for example sheath knockout). If this were possible, such experiment would strengthen the manuscript. Additionally, a comparison of Tse15 cleavage pattern in the background of various mutations could provide interesting insights. For example, could it be that the interaction of Tse15 with VgrG15 is required for autocleavage? Is baseplate or membrane complex formation required for this? If authors have a way to compare the amounts of full-length and processed proteins in various *A. baumannii* strain this could be an interesting dataset.

Reviewer #2 (Remarks to the Author):

In the manuscript titled "Structure of a Rhs effector clade domain identifies new mechanistic insights into type VI secretion system toxin delivery", Hayes, Harper, and colleagues present their findings on the cryoEM structures of a type VI secretion system-associated Rhs effector, Tse15, from *Acinetobacter baumannii*. While the overall molecular architecture of Tse15 appears similar to previously reported Rhs effectors from other species, the authors have identified key differences that distinguish them. Specifically, they have described: 1) the novel structure of the N-terminal clade domain, which has not been resolved previously; and 2) the interacting details between the clade domain and the Rhs cage domain. In order to gain further insights into the interaction between Tse15 and the secretion system, the authors conducted a series of genetic analyses to map the binding regions between Tse15 and its cognate VgrG15 protein. Additionally, the authors reanalyzed their previous secretome data and found supporting evidence for a model in which only the clade and toxin domains of Tse15/16, but not the Rhs cage, are secreted outside of the bacteria to intoxicate adjacent competitor bacteria. Overall, the manuscript is attempting to address the intricate and challenging question of the T6-secretion mechanism. However, it is important to note that the current data regarding the mechanistic insight into toxin delivery is limited, and several key experiments need to be addressed in order to strengthen the manuscript.

Major issue:

1. The lack of validation of the cryo-EM structure is a concern: It is exciting to obtain cryo-EM structures in high resolution. However, it is important to ensure that the structural information accurately reflects the biological reality. i) The authors have identified the amino acid residues involved in the interaction between the clade and the Rhs domains based on their cryo-EM structures. To validate the structure and identify key interacting amino acids, it is recommended that the authors introduce point mutations at the interface and conduct binding assays such as co-immunoprecipitation or analytical size-exclusion chromatography. This would provide direct evidence of the interaction and validate the identified residues. ii) Once the key interacting residues are identified, it would be valuable for the authors to introduce these mutations back into *A. baumannii* Tse15 and conduct T6SS Competitive Killing Assays. This would help validate the importance of the residues and provide further insights into their functional significance in the interaction. (iii) The authors attempted to resolve the unfolded toxin domain by solving two cryo-EM structures (wild type and NN mutant). However, the quality of the resulting structures did not provide sufficient information on how the toxin is embedded in the Rhs cage. This limitation weakens the study's strength as the current data does not support the notion that the density in the cage corresponds to the toxin domain. To support this idea convincingly, the authors could create a toxin-truncated Tse15 variant (1-1389) and demonstrate the absence of density within the cage. Alternatively, the authors could utilize the predicted toxin-Rhs interaction and show that disrupting this binding leads to a malfunction in Tse15 interbacterial killing or toxin delivery. (iv) It is recommended that the authors conduct T6SS Competitive Killing Assays again to demonstrate the importance of the Tse15 residues involved in autocleavage, specifically the D1369N/D1391N, K334A/E343A, and E343A mutations. Addressing these concerns would enhance the validation of the cryo-EM structure, provide further support for the interaction between domains, and increase the understanding of the toxin delivery mechanism in the T6SS system.

2. The evidence to map the interaction between Tse15 and VgrG15 is relatively weak: while using bacterial two-hybrid analysis is a classic and good screening method to identify residues involved in the protein-protein interaction. However, false positive or transient interaction results could lead to misinterpretation and build up incorrect and premature working models. The authors should take advantage of their preliminary genetic data and do tests like co-immunoprecipitation or site-specific crosslinking to obtain conclusive direct evidence of the binding between Tse15 and VgrG15.

3. The idea that the Rhs domain is not exported needs either toning down or more proof: While the concept is interesting, it's hard to imagine how the Rhs cage separates from the clade and toxin during T6 firing. As far as I know, this perspective is new in the field (correct me if I'm wrong). Does the author have evidence that the C-terminal toxin can still attach to the clade or VgrG domain without the Rhs cage? It would be helpful to show this beyond just looking at the secretome? For example, the authors could use a method like immunoblotting to show that the Rhs cage is not presented in the secreted supernatant. There are technical reasons why finding the Rhs domain in the secretome might be tough. To introduce this new idea to the field, the author should provide more proof.

Minor issue:

1. Line 279: There's a typo in "identified."

2. In the manuscript, only +/- signs are used for bacterial two-hybrid data. It's crucial to display the raw images to back up the claims. Two-hybrid data isn't always binary.

3. Figure 6, Model b): The arrow suggests that the Rhs cage domain binds to the clade/toxin domain but not dissociates. I recommend reversing the arrow to avoid confusion about the direction of the Rhs domain.

Point by point response: NCOMMS-23-56051

Title: Structure of a Rhs effector clade domain identifies new mechanistic insights into type VI secretion system toxin delivery.

Reviewer #1 (Remarks to the Author):

In this manuscript, Hayes et al. report cryoEM structure and autoprocessing of Tse15, a T6SS-related toxin from *Acinetobacter baumannii*. The authors show that Tse15 is composed of three domains, one being the Rhs cocoon structure holding a C-terminal toxin domain in an inactive form. Autoprocessing likely releases and activates the toxin. The authors also solved a structure of a mutant where the autoprocessing was blocked by mutation. This provided further insights into packaging of the toxin into the Rhs cocoon. Interestingly, the authors also identified another conserved autocleavage sequence that processes the N-terminal domain independently of the C-terminal processing of the toxin. In addition, the authors identified how Tse15 interacts with its cognate VgrG using a bacterial two-hybrid system. Furthermore, this interaction was suggested to be required for prey killing by the T6SS. Finally, MS analysis suggested that the Rhs cocoon is not secreted out of the cells while both the toxin and the N-terminal domain are. The manuscript is well written and data are clear. I also think that this describes an interesting mechanism and advances our understanding of secretion and activation of T6SS-related Rhs toxins. While I cannot judge all the technical details of the structural analysis, this seems very reasonable.

1. I would however say that primary data, quantification and statistical analysis should be included for all killing assays mentioned in the text (CFU counts).

We have added CFU counts and statistical analysis for the T6SS competitive killing assays as Figure 4b in the revised manuscript.

2. Similarly, for the two-hybrid assays, some examples of how the reporter strain looks on a plate with interacting proteins and with proteins that fail to interact (include positive and negative controls). The authors should provide detailed criteria or examples to illustrate how they differentiated between positive and negative results in their assays.

We have added a description of how we scored the two-hybrid plates in the relevant methods section (Pg 29, lines 943-946 revised track-changed ms) as well as provided a new Figure S14 that shows a photograph of the assay plates for selected alanine mutants. To improve the visual representation of the data in the main manuscript, we have revised Figure 4 and added panel 4c that shows a representation of a positive and no interaction colony.

3. Finally, the last section of the paper should be renamed. The presented data (reanalysis of previous data) support the model that the cocoon is not secreted but say nothing about target cells.

This is an insightful observation raised by reviewer #1. In light of our latest findings indicating the occasional delivery of the Rhs cocoon from the prey cell, we have revised the title to include this finding, and to clarify that our data relates only to secretion and not delivery.

“The Rhs β -cocoon is rarely secreted out of the prey cell.” Page 16, line 518.

4. The authors propose that the Tsi15 immunity protein is required only for sister cell killing. This could be tested as the immunity protein should be possible to delete (in the presence of active Tse15) in T6SS negative strain (for example sheath knockout). If this were possible, such experiment would strengthen the manuscript.

We conducted the experiment suggested and can show that, as we proposed, Tsi15 is only required to prevent sister, but not self-killing. A double mutant was generated that had both the T6SS system inactive (via deletion of *tssM*) and the Tse15 cognate immunity gene *tsi15* deleted. Growth curve analysis showed that the $\Delta tssM tsi15$ double mutant (unable to export Tse15 due to T6SS inactivity) was as viable as wild-type *A. baumannii* AB307. This confirms that Tse15 that is retained in the host is not toxic.

We have incorporated this new data on Pg 16, line 518 of the revised manuscript as the first paragraph of the section titled “The Rhs β -cocoon is most often not secreted out of the prey cell.

“Our structural data clearly show that the Tse15 toxin is retained in the Rhs β -cocoon, suggesting that the cocoon likely protects the host cell from Tse15 toxicity. Thus, we reasoned that the cognate immunity protein Tsi15 is only required to prevent sister, but not self-killing. To test this, we constructed a double mutant that had both the T6SS system inactivated (via deletion of *tssM*) and the *tsi15* gene deleted. This mutant would produce Tse15 but be unable to export Tse15 due to T6SS inactivity and would lack the cognate immunity protein Tsi15 that neutralizes Tse15 toxicity. Growth curve analysis conducted in LB media showed that the $\Delta tssM\Delta tsi15$ double mutant was as viable as wild-type *A. baumannii* AB307 (**Supp Figure 15**), confirming that Tsi15 is not required to protect the cell from self-intoxication with Tse15.”

We have also updated the discussion (Pg 20, line 643) to reflect that we were able to experimentally verify this hypothesis:

“For *A. baumannii* AB307-0294, the Tse15 cognate immunity protein, Tsi15, is encoded upstream of the *tse15* gene. After resolution of the Tse15 structure and with the shielded, unfolded toxin resolved, we predicted that the immunity protein is not required by the host cell that is preparing to fire the T6SS but is rather produced to give immunity to injection of active Tse15 toxin from a neighbouring cell. Indeed, a $\Delta tsi15\Delta tssM$ mutant unable to fire the T6SS and lacking the Tsi15 immunity protein was as viable as wild-type parent AB307_0294, showing that Tsi15 is not required to prevent self-intoxication.”

5. Additionally, a comparison of Tse15 cleavage pattern in the background of various mutations could provide interesting insights. For example, could it be that the interaction of Tse15 with VgrG15 is required for autocleavage? Is baseplate or membrane complex formation required for this? If authors have a way to compare the amounts of full-length and processed proteins in various *A. baumannii* strain this could be an interesting dataset.

We agree that determining the mechanism and requirements of autocleavage is of interest.

We note that to purify recombinant Tse15, we expressed the protein in competent *E. coli* cells. During expression in this non-native cell line, Tse15 was able to undergo autocleavage. Given that VgrG15 and all other components of the T6SS were not present in these cells, it seems highly unlikely that other T6SS proteins play an essential role in Tse15 autocleavage. Instead, we believe it is most likely that protein folding itself initiates autocleavage as it brings the catalytic regions in proximity to their target. This allows autocleavage to occur but does not rely on any external proteins or other factors. Thus, we do not believe further experiments are warranted here.

Text that describes this can be found within the current discussion on page 20, line 658. To further clarify this point we have made the following addition:

“We propose that these events are independent. Both N-terminal clade and toxin self-cleavage occurs following Tse15 main chain protein folding as it brings the catalytic regions in proximity to their target. Following auto-cleavage, the toxin is frequently released from the Rhs cage prior to delivery into the prey cell by the T6SS. This is supported by the fact our secretome data show that the Rhs domain most often dissociates before the firing of the T6SS complex. Further evidence to show that autocleavage of Tse15 (and likely other Rhs proteins) is not dependent on external factors, is that Tse15 autocleavage occurs even when produced in non-native E. coli cells. Finally, we postulate that the Rhs domain itself is not commonly delivered due to the size of the cage as this would effectively ‘blunt’ the end of the T6SS.”

Reviewer #2 (Remarks to the Author):

In the manuscript titled "Structure of a Rhs effector clade domain identifies new mechanistic insights into type VI secretion system toxin delivery", Hayes, Harper, and colleagues present their findings on the cryoEM structures of a type VI secretion system-associated Rhs effector, Tse15, from *Acinetobacter baumannii*. While the overall molecular architecture of Tse15 appears similar to previously reported Rhs effectors from other species, the authors have identified key differences that distinguish them. Specifically, they have described: 1) the novel structure of the N-terminal clade domain, which has not been resolved previously; and 2) the interacting details between the clade domain and the Rhs cage domain. In order to gain further insights into the interaction between Tse15 and the secretion system, the authors conducted a series of genetic analyses to map the binding regions between Tse15 and its cognate VgrG15 protein. Additionally, the authors reanalyzed their previous secretome data and found supporting evidence for a model in which only the clade and toxin domains of Tse15/16, but not the Rhs cage, are secreted outside of the bacteria to intoxicate adjacent competitor bacteria. Overall, the manuscript is attempting to address the intricate and challenging question of the T6-secretion mechanism. However, it is important to note that the current data regarding the mechanistic insight into toxin delivery is limited, and several key experiments need to be addressed in order to strengthen the manuscript.

Major issue:

1. The lack of validation of the cryo-EM structure is a concern: It is exciting to obtain cryo-EM structures in high resolution. However, it is important to ensure that the structural information accurately reflects the biological reality.

i) The authors have identified the amino acid residues involved in the interaction between the clade and the Rhs domains based on their cryo-EM structures. To validate the structure and identify key interacting amino acids, it is recommended that the authors introduce point mutations at the interface and conduct binding assays such as co-immunoprecipitation or analytical size-exclusion chromatography. This would provide direct evidence of the interaction and validate the identified residues.

We only describe what the cryo-EM data shows (Pg 6) and discuss the role of the clade in potential mechanism(s) later in the discussion. It is not a key finding of the paper – simply a description of the data obtained as is generally expected and required of new experimental structures.

The Tse15 cryo-EM structures themselves are validated by the data we present (**Fig S2; Fig S3; Supp Table 1; Fig S7 and Fig S10**). Further data will also be available to the reader via PDB / EMDB codes 8UY4 / 42972 and 8XUT / 42775. Validation reports were also provided for review.

There is a chance with every structure (indeed all laboratory experiments) that artefacts from the technique or approach will influence results. However, our experimental methods and data validation are in line with other published high-quality cryoEM structures. With regard to how well the cryo-EM structure correlates with the biological 'reality' in *A. baumannii*, we feel that our approach is currently using the best practises available to determine structure. Cryo-EM does not impose physical lattice constraints on the protein (as is the case with crystallography) and as such, is often considered to provide a more native view of protein interactions and potential dynamics. The protein frozen on the cryo-EM grids was pure, consisting only of Tse15 that we know is a single species in solution (**Figure 1b**).

With regard to using a mutagenesis approach to validate the identified residues, we suggest that the interface is not a key finding or discovery of this manuscript. We also feel that interruption of the domain interface by extensive mutagenesis would likely cause problems with protein folding. Recombinant Tse15 is produced in a folded and soluble form (that is not toxic to the expression host). We therefore assume protein folding occurs within the single polypeptide prior to auto-cleavage of the three domains, suggesting a complex folding pathway. The interaction interface of the Rhs and clade domains is extensive. There are 15 salt bridges and 19 hydrogen bonds that are produced from a contribution of 34 different Tse15 residues (15 from clade; 19 from Rhs) (**Supplementary Table 3**). Given the likely extensive interface, it is not really viable to identify and mutate 'key interacting amino acids' without significantly altering the protein sequence, surface and subsequent folding. Even if attempted to remove 50% of the salt bridges, we would need to introduce at least 10 different point mutations to a new construct of Tse15 and then attempt to repeat the purification, testing and structure determination. Even if such a mutant could be produced, there is a strong possibility that the remaining residues may still allow the domains to interact.

However, to address the concern we have removed the text (page 6, line 189) that highlights one potential salt bridge to show that we are merely describing overall characteristics, rather than individual interactions.

Edited: "The interaction surface between the clade and Rhs domain shows a total of 19 hydrogen bonds and 15 salt bridges, ~~with one salt bridge located on the interior of the Rhs cage and close to the site of autocleavage (clade residue E326 with Rhs K400)~~ (Supp Table 3, Supp Figure 5d). The high number of potential salt-bridges in the interface suggests that dissociation of the clade would likely be pH labile."

ii) Once the key interacting residues are identified, it would be valuable for the authors to introduce these mutations back into *A. baumannii* Tse15 and conduct T6SS Competitive Killing Assays. This would help validate the importance of the residues and provide further insights into their functional significance in the interaction.

Thank you for this suggestion. To validate the importance of the cleavage regions in Tse15 *in vivo* and provide further insights into their functional significance regarding delivery and toxicity, we made a clade cleavage mutant and a toxin cleavage mutant and assessed their ability to kill *E. coli* prey. This experiment is also suggested by reviewer #2 below in point v) and hence results will be described within that section.

(iii) The authors attempted to resolve the unfolded toxin domain by solving two cryo-EM structures (wild type and NN mutant). However, the quality of the resulting structures did not provide sufficient information on how the toxin is embedded in the Rhs cage. This limitation weakens the study's strength as the current data does not support the notion that the density in the cage corresponds to the toxin domain.

We feel our data does support the conclusion that the toxin is embedded in the Rhs cage and we clearly state for the reader where we only have limited data and have restricted our interpretation accordingly (Pg 6, line 161-166; 176-177; Pg 9, lines 264-269).

We feel the following points provide convincing evidence that the toxin is located inside the Tse15 Rhs cage.

1. One known function of the Rhs cage is to encapsulate the toxin, shielding the host from toxic activity. Rhs effector structures RhsA (*Pseudomonas protegens*, Gunther et al., *Plos Pathog* 2022); Tre23 (*Photorhabdus laumondii*, Jurenas et al., *Nature Comms* 2021) and RhsP (*Vibrio parahaemolyticus*, Tang et al., *Cell Rep* 2022) all conclude that the toxin is located inside the cage despite obtaining less data with more limited density compared to our improved structure.

Similar conclusions were drawn by Donato et al (*PNAS*, 2020) using a molecular microbiology approach to show that the cage encapsulates the toxin in *Enterobacter cloacae* T6SS. These findings were not unexpected as the first ABC toxin that was structurally described also used a Rhs-like cage to encapsulate its toxin (Busby et al., *Nature* 2013).

We have also generated new data for our manuscript that shows that full length Tse15 is not toxic to the prey cell in the absence of its immunity protein and a functional T6SS. This provides biological support that the toxin must be shielded and/or unfolded. This finding correlates well with our structural data. *See also our response to Reviewer 1, point 4.*

2. Our structures were generated from high resolution and high-quality maps that clearly visualises difference density in both datasets after clade and Rhs domains were placed (**Supplementary Figure 6**). Mass spectrometry confirmed that the preparation of Tse15 frozen on grids was pure and no other protein was present. We can confidently place all residues in the clade and Rhs domain, leaving only toxin to be modelled into the remaining density within the cage. The maps have been uploaded to the EMDB (8UY4 / 42972 and 8XUT / 42775) and will be available to the reader upon publication.
3. The NN mutant shows clear, continuous peptide density **of 89 residues of the toxin**. The position of the toxin in both the wild-type and NN mutant is similar, suggesting the NN mutation has not caused catastrophic changes to the protein architecture.

Therefore, we feel there is sufficient data to show that the density observed within the cage is indeed the toxin.

(iv) To support this idea convincingly, the authors could create a toxin-truncated Tse15 variant (1-1389) and demonstrate the absence of density within the cage. Alternatively, the authors could utilize the predicted toxin-Rhs interaction and show that disrupting this binding leads to a malfunction in Tse15 interbacterial killing or toxin delivery.

We feel that generating a truncation mutant cryo-EM structure as suggested will not add any new knowledge to our paper or the field and as such, is an unreasonable request given the high resources required to complete another structure that is unlikely to shed new insight or discovery.

We also have concerns about an extensive mutagenesis approach for the same reasons as outlined in response to point i).

(v) It is recommended that the authors conduct T6SS Competitive Killing Assays again to demonstrate the importance of the Tse15 residues involved in autocleavage, specifically the D1369N/D1391N, K334A/E343A, and E343A mutations. Addressing these concerns would enhance the validation of the cryo-EM structure, provide further support for the interaction between domains, and increase the understanding of the toxin delivery mechanism in the T6SS system.

We thank the reviewer #2 for this suggestion. To address the importance of the clade cleavage motif and the toxin cleavage motif in Tse15, we constructed two marker-less *A. baumannii* genomic mutants by replacing the clade cleavage region or the toxin cleavage region with an insertion encoding an in-frame flippase recognition target (FRT) site. Both mutants grew at rates indistinguishable to the growth of wild-type AB307_0294. Importantly, in competition assays with *E. coli* prey cells vulnerable only to Tse15-mediated killing, both the Tse15 clade mutant and toxin cleavage mutant showed highly significantly reduced ability to kill the *E. coli* prey. Interestingly, the killing ability was not completely abolished. We hypothesize the minimal residual toxicity is due to the T6SS sometimes delivering mutated Tse15 into the prey cell and that subsequent proteolysis of the Rhs cage allows the release of the toxin.

The T6SS competitive killing assay data for the Tse15 clade cleavage and Tse15 toxin cleavage mutants are now included in Figure 1 (panel 1d) and we have added the following lines to the results on Pg 10, line 316.

“To address the biological importance of the two cleavage motifs (clade and toxin) in Tse15, we constructed two marker-less mutants in *A. baumannii* strain AB307_0294 using an in-frame flippase recognition target (FRT) approach. The two mutant strains, *tse15CC::FRT* (AL4734) and *tse15CC::FRT* (AL4745) had the clade cleavage motif (Tse15 amino acids 335-346) or the toxin cleavage region (amino acids 1384-1395) replaced with an in-frame FRT site, respectively. In competition with *E. coli* prey, the Tse15 cleavage mutant strains grew at rates indistinguishable to the growth of wild-type AB307_0294 (**Supp Figure 11d**). Importantly, both had a significantly reduced ability to kill *E. coli* prey vulnerable only to *tse15*-mediated killing (**Figure 1d**). This result indicates that both cleavage sites are important for the proper function of the Tse15 toxin.”

We have also added content to the discussion (Pg 20, line 669) to address the slight, but significant, difference in survival of *E. coli* prey following co-culture with the cleavage mutants compared to the *tse15* deletion mutant.

“Our T6SS competitive killing assay data clearly shows that both the clade cleavage region and the toxin cleavage region are important for Tse15 toxicity. Interestingly, the killing ability of the cleavage mutants (*tse15CC::FRT* and *tse15TC::FRT*) was not completely abolished. We hypothesize that low levels of the mutated Tse15, unable to cleave at either the clade or toxin cleavage site, are delivered into the prey cell by the T6SS system and that subsequent proteolysis of the Rhs cage allows the toxin to be activated. A similar phenomenon has been observed previously for the *Aeromonas dhakensis* T6SS Rhs effector Tse1. Recombinant Tse1 is toxic when expressed in *E. coli* and although the protein was significantly reduced in toxicity when the clade cleavage region or toxin cleavage region was mutated it still retained some residual activity²⁷.”

2. The evidence to map the interaction between Tse15 and VgrG15 is relatively weak: while using bacterial two-hybrid analysis is a classic and good screening method to identify residues involved in the protein-protein interaction. However, false positive or transient interaction results could lead to misinterpretation and build up incorrect and premature working models. The authors should take advantage of their preliminary genetic data and do tests like co-immunoprecipitation or site-specific crosslinking to obtain conclusive direct evidence of the binding between Tse15 and VgrG15.

We thank the reviewer for this suggestion and agree that this significantly improves the paper. We conducted the suggested co-immunoprecipitation experiment using recombinant proteins and indeed show that Tse15 and VgrG15 do interact, thus supporting our bacterial two-hybrid analyses. To perform this experiment, we bound purified recombinant Tse15 to nickel beads and washed over the cellular supernatant of *A. baumannii* AB307 to identify binding partners. As the VgrG proteins are not produced at high levels natively, we used lysate from a strain that expressed VgrG15 both from the chromosome and from a recombinant plasmid and also had the other *vgrG* genes (*vgrG16* and *vgrG17*) deleted, so that all expression of VgrG proteins was VgrG15 specific. Following the pulldown, we conducted peptide fingerprinting analysis of all identified protein bands. We were able to identify VgrG15 as well as a small number of other proteins. Importantly, VgrG15 was recovered from the same excised band that also contained Tse15. These data confirm that Tse15 and VgrG15 are indeed interaction partners. This experiment has been added to the manuscript, with the resulting pulldown gel and mass spectrometry results added to the supplementary material (**Supp Figure 12**). The text has been also edited to reflect the confirmation of these findings (Pg 11, line 333):

“In *A. baumannii* strain AB307-0294, delivery of Tse15 into *E. coli* prey cells requires a functional VgrG15 protein. This was initially determined by Fitzsimons et al. and was experimentally confirmed in our study by a pulldown of native VgrG15 by purified Tse15 (**Supp Figure 12**). To map the specifics of the interaction between VgrG15 and Tse15, we...”

3. The idea that the Rhs domain is not exported needs either toning down or more proof: While the concept is interesting, it's hard to imagine how the Rhs cage separates from the clade and toxin during T6 firing. As far as I know, this perspective is new in the field (correct me if I'm wrong). Does the author have evidence that the C-terminal toxin can still attach to the clade or VgrG domain without the Rhs cage? It would be helpful to show this beyond just looking at the secretome? For example, the authors could use a method like immunoblotting to show that the Rhs cage is not presented in the secreted supernatant. There are technical reasons why finding the Rhs domain in the secretome might be tough. To introduce this new idea to the field, the author should provide more proof.

Reviewer #2 is correct in stating that this is a novel concept to the field, and as such we have conducted further experiments to further validate these findings. We again utilised secretome analysis, but we also assessed what proteins were able to be identified in the whole cell lysate of the same *A. baumannii* AB307 cells. This allows a point of comparison and can confirm the lack of cage peptides in the secretome does not result due to technical reasons. Given that AB307 produces two Rhs proteins, Tse15 the main focus of this paper, and Tde16, a DNase, we have measured and reported cage peptides for both of these proteins. We believe that this strengthens and broadens the conclusions.

In the whole-cell fraction, we were able to identify a very large number of cage-specific peptides for both Tse15 and Tde16, indicating that the MS can robustly identify these peptides. However, there were significantly reduced Rhs cage peptides of Tse15 in the secreted fraction, and very few cage peptides of Tde16. This shows that indeed there is a marked reduction in cage peptides outside of the cell, and the presence of Rhs cage peptides in the whole cell proves that this finding is not due to technical difficulties. We note that during this experiment we were able to identify a greater number of secreted peptides than in both of our previous secretome experiments, which is likely due to the increased sensitivity of the MS hardware used for this new experiment. Given this, we have modified

the text to state that the Rhs cage can be occasionally delivered with the toxin, but this is most often not the case. We have added the new data as Figure 6 and Table S5 as well as modified the text as outlined below to reflect these findings:

Abstract (Pg 2)

“Proteomic analyses showed that a larger number of peptides from the Tse15 N-terminal clade and toxin domains, compared to the beta-strand rich Rhs cage, are secreted outside of the cell, suggesting a novel mechanism for Rhs toxin delivery and activation. Our findings suggest that this delivery mechanism requires an interaction between the N-terminal clade and toxin domains, with the clade domain acting as the internal chaperone to mediate tethering of the toxin to the T6SS machinery.”

Results (new paragraph, new Fig 6, new Table S5, Pg 17 -18)

To confirm the highly reduced delivery of Tse15 (and Tde16) Rhs cage peptides, we repeated the proteomics with the addition of analysis of whole-cell lysate samples (**Figure 6**). As expected, in the whole-cell lysate samples, peptides covering the majority of Tse15 and Tde16 were identified. In the supernatant samples, we identified increased numbers (0.9-1.6-fold) and intensity (2.2-7.6-fold) of peptides for the clade and toxin regions but highly reduced peptides (0.04-0.09-fold) and intensities (.07-.34-fold) for the cage peptides for both Tse15 and Tde16 (**Supp Table 5**). By comparison, for two other T6SS proteins (Hcp and Tae17) peptide coverage and intensities for peptides in the supernatants were increased across the whole proteins (1-2.3-fold for unique peptides and 4.1-6.5-fold for peptide intensities). RpoB was also analysed as a non-secreted cytoplasmic protein; whole-cell lysate samples showed coverage across the whole proteins, while very few peptides were observed in the supernatants samples, indicating the samples displayed very low levels of sample lysis. The values for RpoB in the supernatant are similar to that observed for the Tse15 and Tde16 cage, suggesting that the cage peptides observed may also result from sample lysis, rather than direct secretion. Overall, the two independent experiments confirm that for both Tse15 and Tde16, the clade and toxin domains are secreted from the cell, but the cage is retained in the host.

Discussion

“This is supported by the fact our secretome data show that the Rhs domain most often dissociates before the firing of the T6SS complex. We postulate that the Rhs domain itself is commonly not delivered due to the size of the cage as this would effectively ‘blunt’ the end of the T6SS.” (Pg 20)

“Taken together, our data allow us to present a novel Rhs effector delivery mechanism. The Rhs protein is first responsible for delivering the toxin to the VgrG, with Tse15:VgrG15 interactions involving both the N-terminal clade and Rhs domains of Tse15. Once Tse15 is bound to VgrG15, we postulate that either the Rhs domain may dissociate immediately, is retained inside the cell when the system fires or infrequently can still be fired from the prey cell (**Figure 6a**). In the first two scenarios, a large conformational shift must occur either before or during firing. This conformational shift may be mediated by changes to salt bridges across the N-terminal clade and Rhs domain interface. Importantly, interactions between the clade domain and the toxin CTD within the Rhs cage allow these domains to remain associated if the Rhs domain dissociates.” (Pg 22)

“We have also mapped the interactions of Tse15 with its cognate VgrG15, and secretome data show that the Rhs cage of both AB307-0294 Rhs proteins (Tse15 and Tde16) are often not secreted outside of the predator cell. Thus, we propose that the Rhs domain of Tse15 is essential for delivering the effector to VgrG15 and initiating the binding interaction. However, the Rhs domain most often then dissociates or stays associated until firing, upon which physical interactions with the predator cell membrane and/or T6SS membrane structures forces the Rhs domain to dissociate. These events allow the N-terminal clade domain together with the CTD toxin domain to exit the predator cell. This model is in contrast to all previous hypotheses that entire Rhs effectors are always delivered into the prey cell³².” (Pg 23)

“Finally, the fact that the Rhs cage is commonly left behind fundamentally changes our understanding of toxin release and how future proteins may be engineered for delivery of novel T6SS payloads.” (Pg 23)

Minor issue:

1. Line 279: There's a typo in "identified."

This typo has been resolved.

2. In the manuscript, only +/- signs are used for bacterial two-hybrid data. It's crucial to display the raw images to back up the claims. Two-hybrid data isn't always binary.

This point was also raised by reviewer #1 and has been addressed. See response to Reviewer 1, point 2.

3. Figure 6, Model b): The arrow suggests that the Rhs cage domain binds to the clade/toxin domain but not dissociates. I recommend reversing the arrow to avoid confusion about the direction of the Rhs domain.

We have corrected the figure in the revised manuscript.

REVIEWER COMMENTS

Reviewer #1 (Remarks to the Author):

Thank you for addressing my comments!

Reviewer #2 (Remarks to the Author):

The revised manuscript addresses some of my concerns, but not all. One significant novelty of the manuscript is the resolution of the N-terminal clade domain and the demonstration of interaction between the clade and rhs domains, a previously unreported finding. The discovery of this interaction is intriguing and could potentially contribute to the mechanistic understanding of cargo effector export via the type VI secretion system. The authors suggest that mutagenic analysis was not feasible due to potential protein folding issues; however, they have not provided additional analytic data to support this claim. I disagree with their rebuttal statement that "It is not a key finding of the paper – simply a description of the data obtained as is generally expected and required of new experimental structures." If the authors do not consider it a key finding, the rationale for resolving the structure and presenting the data in the manuscript is unclear. Therefore, I strongly recommend conducting mutagenesis analyses along with secretion/competitive killing assays to identify important interacting residues between the clade domain and the Rhs cage, thereby elucidating the biological significance of the interaction.

The second concern is the mapping of the interaction between VgrG and Tse15. The authors conducted pull-down MS to demonstrate that Tse15 Δ tox binds to VgrG from the lysate. This experiment should be supplemented with a control using Tse15 Δ tox without amino acids 79-133 or 542-570. This approach would provide stronger evidence than two-hybrid assays that these segments (79-133 and 542-570) are indeed crucial for the binding between the two proteins. Additionally, given the authors' genetic evidence indicating key residues/segments on VgrG that interact with Tse15, they should utilize their established pull-down MS to verify the importance of these interacting residues.

Finally, although the authors have shown MS data indicating that the number of RHS peptides in the supernatant is relatively low compared to the whole-cell sample, the presence of RHS peptides from Tse15 (amino acids 133-1389) cannot be overlooked. I recommend that the authors provide more evidence to support their proposed model. It is challenging to envision how the Rhs cage separates from the clade and toxin during T6SS firing. The authors should demonstrate whether the C-terminal toxin can still attach to the clade or VgrG domain in the absence of the Rhs cage.

Point by point response: NCOMMS-23-56051B

Title: Structure of a Rhs effector clade domain identifies new mechanistic insights into type VI secretion system toxin delivery.

Reviewer #1 (Remarks to the Author):

Thank you for addressing my comments!

Reviewer #2 (Remarks to the Author):

The revised manuscript addresses some of my concerns, but not all.

One significant novelty of the manuscript is the resolution of the N-terminal clade domain and the demonstration of interaction between the clade and rhs domains, a previously unreported finding. The discovery of this interaction is intriguing and could potentially contribute to the mechanistic understanding of cargo effector export via the type VI secretion system. The authors suggest that mutagenic analysis was not feasible due to potential protein folding issues; however, they have not provided additional analytic data to support this claim. I disagree with their rebuttal statement that “It is not a key finding of the paper – simply a description of the data obtained as is generally expected and required of new experimental structures.” If the authors do not consider it a key finding, the rationale for resolving the structure and presenting the data in the manuscript is unclear. Therefore, I strongly recommend conducting mutagenesis analyses along with secretion/competitive killing assays to identify important interacting residues between the clade domain and the Rhs cage, thereby elucidating the biological significance of the interaction.

RESPONSE: We fully agree with the reviewer that an important novelty of the work is the resolution of the clade domain; this is the first time that such a domain has been structurally determined and additionally we have shown that this fold is likely commonly found across numerous type VI secretion system effectors. This information will be of major interest to many researchers working with similar systems. Our cryo-EM structure clearly identifies the fold of the clade domain, and a likely interaction interface between the clade and Rhs domains. However, we do not infer specific functionality of the interaction between clade and Rhs – but describe in detail how the cryo-EM structural data indicates this interaction. Such a description of the data is essential for any new structural publication. We stated in our previous response that we do not see the *interface residues* as a main conclusion of this paper, but the reviewer appears to have confused this statement with the idea that we do not think that resolution of the clade domain is an important conclusion (which it is).

With respect to further work to fully define the Clade:Rhs interaction interface, we do not believe that a mutagenesis approach, as outlined by reviewer 2, would achieve this. The interaction surface is large (buried surface area is 2396 Å²) and engages residues located on four different secondary structure elements in the clade (3 helices, 1 strand) and residues spread over 8 separate β-strands from the Rhs domain. The length of this interaction interface is ~ 28 Å, and there is no focal or central area that dominates the interactions. Therefore, this is a very extensive interface with a broad region of crucial interactions (15 salt bridges & 19 hydrogen bonds). Individual amino acid mutations are highly unlikely to disrupt this interface, while more significant internal deletions are very likely to disrupt the overall fold of either domain, making biologically relevant interpretation problematic.

However, we have added some additional information into the text to more clearly explain this to the reader, noting why mutagenesis was not attempted. (Pg5, line 164, revised manuscript)

“The interaction surface *between the clade and Rhs domain was long, spanning 8 β-strands of the Rhs cage (~28 Å) and had a buried surface area of 2396 Å². Interrogation of the residue interactions* between the clade and Rhs domain showed a total of 19 hydrogen bonds and 15 salt bridges (Supp Table 3, Supp Figure 5d). *Due to the spread of these residues, we did not pursue a mutagenesis approach to verify the interaction surface as individual amino acid mutations were*

unlikely to disrupt the interface, while more significant internal deletions were likely to disrupt the overall fold of either domain. The high number of potential salt-bridges in the interface does, however, suggest that dissociation of the clade would likely be pH labile.”

Reviewer 2 Point 2. The second concern is the mapping of the interaction between VgrG and Tse15. The authors conducted pull-down MS to demonstrate that Tse15 Δ tox binds to VgrG from the lysate. This experiment should be supplemented with a control using Tse15 Δ tox without amino acids 79-133 or 542-570. This approach would provide stronger evidence than two-hybrid assays that these segments (79-133 and 542-570) are indeed crucial for the binding between the two proteins. Additionally, given the authors' genetic evidence indicating key residues/segments on VgrG that interact with Tse15, they should utilize their established pull-down MS to verify the importance of these interacting residues.

RESPONSE: At the request of Reviewer 2 in their first review, and previously added into revision 1 of the manuscript, we provided significant further data that clearly supports the direct interaction of native VgrG15 and recombinant Tse15, and we showed by mutagenesis and biological assays the importance of the clade and toxin cleavage sequences. The Tse15 interacting regions identified by two-hybrid (shown in Fig 3b in red in revised manuscript) are integral to the fold of Tse15. In our opinion, deleting these internal regions from the protein, as suggested here by reviewer 2, would almost certainly disrupt the overall fold of the protein and raise more concerns on biological relevance than they would answer. If the protein cannot fold similarly to the native protein, then these functional studies would not be informative. In support of our contention on disrupted folding, we have used AlphaFold3 to assess the Tse15 structure with and without the internal deletions suggested by reviewer 2.

In summary,

1. When AlphaFold3 is used to model the wildtype Tse15 Δ tox structure (without internal toxin sequence as we have used for several experiments) the top five models match our cryoEM structure very closely and are also nearly identical with each other (RMSD 0.5-1.6 Å). Therefore, we have high confidence in the AlphaFold3 model accuracy.
2. The top five AlphaFold3 models for the Tse15 Δ 79-133 (as suggested by reviewer 2) are consistent with each other but differ markedly from the wildtype Tse15 models (and therefore our CryoEM structure). The position of the clade domain is completely disrupted and as such we contend that any studies on this mutant would not be biologically relevant (Response Fig. 1A, Pg 4).
3. The top five AlphaFold3 models for the Tse15 Δ 542-570 (as suggested by reviewer 2) are also consistent with each other but have a very disrupted position of the top part of the RHS cage compared to wildtype Tse15 (Response Fig. 1B). Again, we contend that any studies on this mutant would not be biologically relevant
4. By comparison, AlphaFold3 models of the Tse15 with the clade substitution mutation (Tse15CC:335-346, as added previously into the updated manuscript) where we changed amino acids at the clade cleavage site and showed its biological importance by competitive killing assays, are nearly identical to the wildtype Tse15 and are therefore highly biologically relevant (Response Fig. 1C)

Reviewer 2 Point 3. Finally, although the authors have shown MS data indicating that the number of RHS peptides in the supernatant is relatively low compared to the whole-cell sample, the presence of RHS peptides from Tse15 (amino acids 133-1389) cannot be overlooked. I recommend that the authors provide more evidence to support their proposed model.

RESPONSE: Reviewer 2 agrees that our new additional data does support our claims. In the updated manuscript we have provided two separate MS datasets and for two different Rhs proteins (Tse15 and Tde16). The reviewer is correct that in the newer dataset we did identify some secreted peptides from the Rhs cage. However, our data clearly show that the peptides from the Rhs region are present in very high amounts in the whole cell lysates but significantly reduced (Tse15) or almost completely absent (Tde16) in the supernatant. Additionally, we have added in new data showing that there is no difference in measured peptides in whole cell lysates and

supernatant for the third non-Rhs type effector (Tae17). Thus, the observed phenomenon of reduced peptides in the supernatant for a specific effector domain (cage peptides) is specific for Rhs type effectors. This is very exciting and novel data and will be of great interest to all those working on Rhs proteins. We have also added an analysis of peptides from the cytoplasmic protein RpoB, showing that a very small number of these peptides are also observed in the supernatant, suggesting that there is a very low (but measurable) level of lysis in the samples. We strongly contend that this MS data is the most appropriate method to show that the Rhs cage is primarily retained inside the cell following firing.

However, we have modified the text slightly to take account of the reviewers concerns and explicitly note that it is likely that in a low number of T6SS firings, the Tse15 cage is secreted (Pg 14, line 447-453, revised manuscript).

“The values for RpoB in the supernatant are similar to that observed for the Tde16 cage, suggesting that the cage peptides observed for Tde16 may be primarily the result of low-level sample lysis, rather than direct secretion. For Tse15, it is likely that the cage is most commonly retained within the cell but may be secreted at very low levels. Overall, the two independent experiments show that for both Tse15 and Tde16, the clade and toxin domains are readily secreted, but the cage is most often retained within the host cell.

Reviewer 2 Point 4. It is challenging to envision how the Rhs cage separates from the clade and toxin during T6SS firing. The authors should demonstrate whether the C-terminal toxin can still attach to the clade or VgrG domain in the absence of the Rhs cage.

RESPONSE: As we (and others), have clearly shown that the clade, toxin and Rhs cage are auto cleaved (and we have also directly shown the biological relevance of these cleavage positions in the updated manuscript), we do not believe that the idea of separation of Rhs cage from clade and toxin is totally surprising (or “challenging”). We note that the cryo-EM structure shows that the clade region “reaches” inside the cage, in a manner that would allow direct interaction with the toxin domain.

We have added a reference to Supp Fig 5c on Pg 7, line 207 to direct the reader to the figure that shows the proximity of the clade and toxin backbone.

“Also localised close to the entrance and chain C was the longer chain E (43 residues) that appeared to interact with the end of the clade domain, close to the cleavage site (**Supp Fig 5c**). The position of the toxin is intriguing and points to a possible role for the clade in toxin release.”

We do not believe that producing the domains in isolation (toxin, clade and VgrG) would give information on the interaction in a way that would accurately reflect the biology of the system, as our cryo-EM data clearly show that the Rhs cage is fundamental to the positioning of the three regions and therefore the mechanism of interaction. Furthermore, the toxin domain is unfolded inside the Rhs cage, which is unlikely to be the case if the toxin were to be expressed without the other components (indeed our previous work shows that expression of this domain alone gives a full active (folded) toxin (doi: [10.1128/IAI.00297-18](https://doi.org/10.1128/IAI.00297-18)). Thus, we believe that removing the Rhs cage would completely alter the biological system and, in our opinion, not provide biologically relevant insight or knowledge.

Please see Response Figure 1 on next page (Pg 4).

Additional Expert Review:

The authors note that the predicted structure is low confidence for the CTD and so not included in the modeling (lines 124-125). However subsequent figures clearly include "the toxin domain (in) blue" (figure 2 most notably, but also Fig 3 and Supp 5. Do the authors mean that they only included the residues for which they had sufficient density to model independently from the map, and that they did not use the predicted structure for this domain? That's my interpretation from reading the whole thing but I think the authors should state this more clearly at this early point.

A couple of minor point of clarification

- the text in lines 143-146 notes that Supplemental Figure 4 uses structural predictions from AlphaFold but the figure legend does not make this clear.
- Line 164 has a "b" where I think it's meant to be a "beta"

Point by point response: NCOMMS-23-56051B (C?)

Title: Structure of a Rhs effector clade domain identifies new mechanistic insights into type VI secretion system toxin delivery.

Additional expert review:

The authors note that the predicted structure is low confidence for the CTD and so not included in the modeling (lines 124-125). However subsequent figures clearly include “the toxin domain (in blue)” (figure 2 most notably, but also Fig 3 and Supp 5. Do the authors mean that they only included the residues for which they had sufficient density to model independently from the map, and that they did not use the predicted structure for this domain? That’s my interpretation from reading the whole thing but I think the authors should state this more clearly at this early point.

Yes, the reviewer is correct. The toxin domain prediction by AlphaFold was poor and is only shown in Supplementary Figure 3. The density and cartoons of the toxin domain shown in Figures 2, 3 and Supp 5 depict the experimental data we were able to model. To make this clear to the reader, we have added the following

We added the following to line 149 (word doc)

“The top ranked AlphaFold2 model without the toxin domain (residues 1 – 1395) was fitted to the map using UCSF Chimera fitmap.”

Added to Figure legend 2

Figure 2. Single particle cryoEM structures of Tse15. a) Tse15 wild-type density map at a threshold of 0.75 (2.85 RMSD) where the N-terminal domain is coloured orange, Rhs cage grey and toxin peptide(s) in blue. **b)** Cartoon depiction of Tse15 structure coloured by domain (clade orange, Rhs grey and toxin peptides blue). Four separate, unsequenced toxin peptides (106 of 195 residue CTD domain) could be modelled into the density map.

A couple of minor point of clarification

- the text in lines 143-146 notes that Supplemental Figure 4 uses structural predictions from AlphaFold but the figure legend does not make this clear.

We have added this information into the Supplemental Figure 4 legend.

- Line 164 has a “b” where I think it’s meant to be a “beta”

Corrected